# Predicting bacterial phenotypic traits through improved machine learning using high-quality, curated datasets
Julia Koblitz [1] ✉, Lorenz Christian Reimer[1], Rüdiger Pukall[1] & Jörg Overmann [1,2]

Predicting prokaryotic phenotypes—observable traits that govern functionality, adaptability, and interactions—holds significant potential for fields such as biotechnology, environmental sciences, and evolutionary biology. In this study, we leverage machine learning to explore the relationship between prokaryotic genotypes and phenotypes. Utilizing the highly standardized datasets in the BacDive database, we model eight physiological properties based on protein family inventories, evaluate model performance using multiple metrics, and examine the biological implications of our predictions. The high confidence values achieved underscore the importance of data quality and quantity for reliably inferring bacterial phenotypes. Our approach generates 50,396 completely new datapoints for 15,938 strains, now openly available in the BacDive database, thereby enriching existing phenotypic resources and enabling further research. The open-source software we provide can be readily applied to other datasets, such as those from metagenomic studies, and to various applications, including assessing the potential of soil bacteria for bioremediation.

Despite significant advances in molecular microbiology, our understanding of the biology of microorganisms, particularly of prokaryotes, remains incomplete due to the pronounced imbalance between the abundance of genomic data and the scarcity of phenotypic data.

Genome sequencing has become a routine practice thanks to technological advancements, making it faster and more affordable to sequence entire genomes. The genome sequences obtained are essential for the identification and classification of prokaryotes, and are meanwhile mandatory for valid species descriptions[1]. According to a recent proposal, prokaryotic taxa could even be named solely on the basis of metagenome-assembled genomes (MAGs) and in the absence of cultured representatives and their phenotypic information[2], which, however, may lead to incomplete or misleading classifications[3].

While genome sequences provide the basis for such analysis of prokaryotic diversity, genomic data often fall short of providing functional information as many genes present in prokaryotic genomes remain unannotated or poorly annotated. For example, understudied bacterial phyla have an average annotation level of only 44.8%, while even the much better studied *Pseudomonadota* (previously Proteobacteria) and *Bacillota* (previously Firmicutes) still reach only about 57.4% on average[4]. Especially in non-model organisms this impedes the inference of phenotypic traits from genomic data. So far, phenotypic data are almost entirely collected from cultivated bacteria, which introduces strong biases and limits our comprehension of functional microbial diversity, as only about 20,000 species have been validly described[5], whereas estimates of global prokaryotic species numbers range from 800,000 to 1 trillion, with a billion being the most realistic approximation[6–8]. In addition, obtaining phenotypic data is often more time-consuming and costly than sequencing[9]. As a result, our understanding of the role of microorganisms in biogeochemical processes such as nutrient cycling in agricultural soils and greenhouse gas emissions, or the search for new active substances is limited[10]. Addressing these knowledge gaps is also crucial for advancing biotechnology and medical microbiology, where phenotypic traits like antibiotic resistance and metabolic capabilities have direct implications.

The imbalance between the amount of genome sequences and phenotypic data is even apparent for the isolated strains themselves as documented by the contents of the BacDive database. BacDive is the world's largest open database of strain-level phenotypic data, covering physiological data for 99,392 strains of 21,578 species[11] and has been acknowledged as a Global Core Biodata Resource and as ELIXIR Core Data Resource, documenting its comprehensiveness and actuality. While 70% of the 21,168 type strains in BacDive have genome sequences, phenotypic data coverage varies widely (Fig. 1). For instance, data on the response to Gram-staining, a basic phenotypic test, are available for only half of the type strains. If applied to the whole dataset and not limited to type strains, this data gap is even more obvious with only about 17% of all strains having information on

[1]Leibniz Institute DSMZ-German Collection of Microorganisms and Cell Cultures, Braunschweig, Germany. [2]Technical University of Braunschweig, Institute for Microbiology, Braunschweig, Germany. ✉e-mail: julia.koblitz@dsmz.de

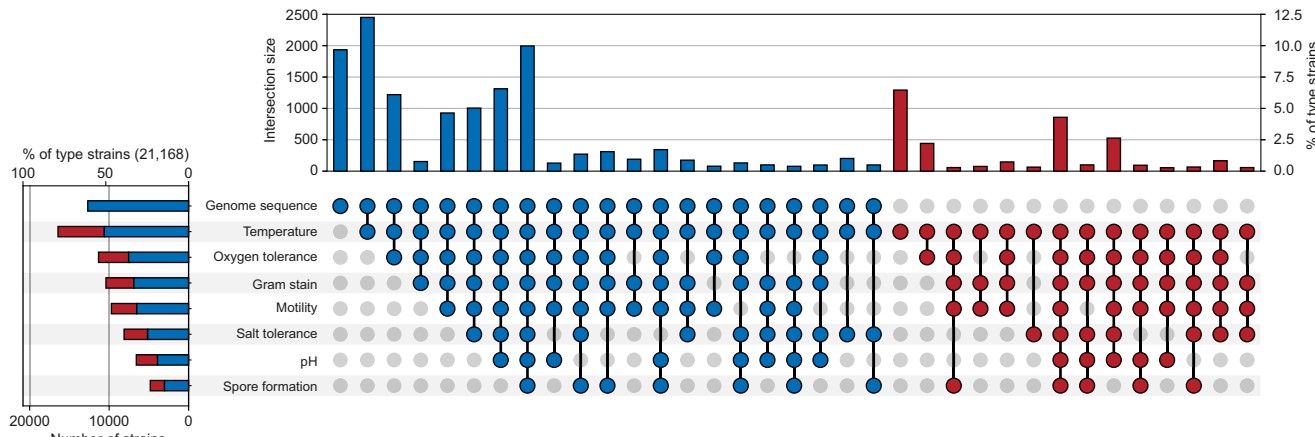

**Fig. 1 | Availability of genome sequences and most common types of phenotypic information for the bacterial type strains covered by the Bac*Dive* database.** Data points in the UpSet plot are linked when more than 50 strains are available for the combination of traits. Records are colored blue if a genome is present and red if not, which is also summarized for traits on the left. A summary of all strains regardless of type strain status is depicted in the Supplementary Fig. 1.

Gram-staining (Supplementary Fig. S1). Finally, the number of prokaryotic species that is recognized through their genomes or MAGs only (113,104 species clusters[12]) already exceeds the number of cultivated bacteria by almost sixfold. Taken together, the large number of available genome sequences provides a huge and growing potential for the inference of microbial functions that are still not known.

Phenotypic traits can be predicted from genomic data using various artificial intelligence (AI) approaches. Early proof-of-concept studies and command-line tools demonstrated the feasibility but were limited in scalability. For example, decision trees and support vector machines were used to predict phenotypic properties, showing promising results but limited to specific datasets and traits[13]. The PICA framework, a command-line tool for predicting phenotypes from genomic sequences, provided a foundation for automated predictions[14], and was integrated into the database PhenDB, which increased accessibility. Another web-based tool named Traitar was created to predict pathogenicity and antibiotic resistance, demonstrating the utility of such platforms in the biomedical context[15]. However, these algorithms and databases are rarely updated; for example, PhenDB had its last update in 2020 and Traitar is not available online anymore. Additionally, many of these tools operate at the species level and not at the strain level, even though strains of the same species can show different phenotypes[16,17]. Model-based studies showcased predictive power but faced issues with overfitting and interpretability, which renders their application to novel datasets less reliable. Jaillard and colleagues demonstrated the potential of machine learning models for generalizing across diverse bacterial taxa, yet these models often exhibited biases towards well-studied model organisms, limiting their applicability to the less-studied taxa[18]. Certain approaches to predict specific phenotypes like antibiotic resistance or carbon source utilization[19], were highly accurate for the target traits, but could not be applied readily to many other microbial traits. In conclusion, current AI approaches exhibit a strong taxonomic bias, are based on incomplete functional annotations, and often deal with noisy and limited data, which diminishes the reliability of the predictions made[20].

Machine learning approaches vary significantly in complexity, interpretability, and predictive performance. While ensemble methods such as gradient boosting algorithms or neural networks may offer superior accuracy, they often sacrifice interpretability and require extensive computational resources[21]. Unsupervised methods, though useful for exploratory analysis, are generally less suited for targeted phenotype predictions due to their lack of labeled guidance[22]. Thus, our choice of Random Forest balances predictive robustness with easy biological interpretability, crucial for extracting meaningful insights from genomic data.

In this study, machine learning was employed to predict phenotypic traits from genomic data on the strain-level. Overcoming several limitations of current tools for phenotype predictions, our approach capitalizes on the high-quality, standardized training datasets that have become accessible through the Bac*Dive* database, is able to incorporate genes without functional annotation, and employs the Pfam annotations of protein families. We showcase several machine learning models for predicting specific traits, discuss the evaluation metrics, and explore the biological implications of our models. The models with the best performance were used to enrich the BacDive database, thereby enhancing the data foundation for future microbiological research. The models are provided as open-source software, allowing future applications to other datasets.

## Results and discussion

The Bac*Dive* database currently (2024.2 release) comprises more than 2.7 million data points covering more than 1000 data fields[23]. However, data availability varies markedly for the different traits and different taxa. In addition, genome sequences are not yet available for all strains in the database. Therefore, we selected traits for machine learning and downstream modeling if the corresponding genome sequences and the states of traits were available for enough strains. For this study we only selected traits with available data for more than 3000 strains. However, as Random Forest is a robust algorithm that can handle a large quantity of features such as protein families[24], it might be possible to lower the threshold to 500–1000 strains per trait in the future to predict traits that are less available in the literature.

Only a limited number of the selected traits, such as the response to Gram-staining, could be classified into two states (i.e., Gram-positive and Gram-negative, ignoring the small number of Gram-variable strains). In contrast, a considerable fraction of prokaryotic traits can assume more than two states as is the case for oxygen tolerance, or are even continuous, as the temperature dependence of growth. We therefore devised and tested approaches that render data from prokaryotic traits with multiple states amenable to machine learning. In parallel, we assessed the effects of an uneven distribution of data points among the different trait states on the quality of predictions (determined by different metrics). From the models obtained, the importance of individual features (i.e., protein families) was deduced to arrive at a biological interpretation of the machine learning model.

### Evaluation and justification of protein annotation methods and modeling approaches

In this study, we chose protein family annotations of the Pfam database as the basis for our machine learning approach after carefully evaluating alternative genomic annotation methods. Pfam is widely regarded as a comprehensive and reliable source for annotating protein domains and

families due to its extensive coverage, clearly defined domain boundaries, and frequent updates[25]. Furthermore Pfam has a much higher mean annotation coverage (80%[25]) than Prokka (52%[4]). We benchmarked Pfam annotations against other annotation tools, including eggNOG, and other tools of InterPro such as SMART, PRINTS, SUPERFAMILY, and CDD[26,27]. While eggNOG provided equally strong predictive performance, it required significantly longer computational runtimes and produced an excessively high number of distinct functional categories—a test annotation with 3000 bacterial genomes generated more than 2 million distinct ENOG orthologous groups, complicating downstream analysis. Tools such as SMART and PRINTS were limited by their relatively small number of classes, negatively affecting model performance. Conversely, SUPERFAMILY annotations were overly condensed, merging many distinct functional classes and thereby reducing the discriminative power of the resulting models. Although CDD[28] represented a viable alternative, we selected Pfam due to its optimal balance between granularity and interpretability, alongside its established usage in the literature.

We also considered modern transformer-based genomic sequence models, which have shown promising results in numerous genomic prediction tasks and are mostly applied to human genomes so far[29,30]. Despite their potential, these models are particularly sensitive to subtle, potentially irrelevant genomic variations such as single nucleotide polymorphisms (SNPs) or non-coding regions. Recent literature has highlighted that transformer models, due to their highly parameterized nature and their capability to capture fine-grained sequence patterns, can inadvertently develop strong taxonomic biases and are furthermore challenging to interpret[21]. They also need significant computational power and a substantial amount of training data which cannot be provided for many traits[31,32]. Although transformer-based methods hold considerable promise for genomic analyses, we chose a feature-based machine learning approach utilizing Pfam annotations, not only to mitigate possible biases but also to allow for clearer biological interpretability, which was very important to the conclusions of this study.

## Beyond binary states of traits: predicting oxygen requirements

With regard to oxygen requirements, prokaryotic strains are typically classified as anaerobes, facultative anaerobes, aerobes, aerotolerant, or microaerophilic species, depending on the amount of oxygen required or tolerated by the organism. For such traits with non-binary states, a multi-label classification model could in principle be built and trained to distinguish between each of these classes. However, this would only yield reasonable results if the data points were rather evenly distributed between the multiple classes and if individual classes could be well discriminated from each other. Both conditions are not met for data describing the oxygen demand of a prokaryotic strain. For instance, contradictory data from literature are apparent in the BacDive database, where 7.1% of the strains have more than one state listed for their oxygen requirement (Fig. 2A; data points linked by horizontal lines).

To meet this challenge, two separate models for oxygen tolerance were trained: one for aerobic (AEROBE) and one for anaerobic lifestyle (ANAEROBE). Intermediate cases, i.e. facultatively anaerobic or aerobic, or microaerophilic, were treated as negative while ambiguous datasets were removed from the training and test set. The resulting predictions can be expressed as a vector with 1 for positive and 0 for negative for both models. The underlying assumption is that the models correctly identify strains that are either aerobes [1 0] or anaerobes [0 1] with high confidence, while failing to predict facultative anaerobes and microaerophiles. This failure can be expressed as [0 0] (both models negative), where the assumption is that they are either facultative anaerobe or others. The cases where both models predict positive [1 1] serve as a control for the quality of the models, since one of them must be wrong.

For the test dataset, the AEROBE model performed well in predicting aerobic strains with a precision of 97.0%. However, the recall was remarkably lower with only 87.9% of the truly aerobic strains found among the test data, hence the number of false negatives was higher than that of false

positives. This yielded an F1 score of 92.3. On the opposite, the ANAEROBE model had a lower precision of 88.7% while almost all true anaerobes were identified with a recall of 98.2%.

To evaluate how well the two models separate the two trait states, the confidences of the AEROBE model were compared to the confidences of the ANAEROBE model (Fig. 2B). The 2D kernel density plot revealed that the two models successfully determined either true positives of aerobes or of anaerobes with marginal overlap. In fact, only one single test strain was simultaneously predicted to be positive by the AEROBE and the ANAEROBE model (with a confidence of 61% and 54%, respectively). Accordingly, the application of the two models allowed to clearly distinguish between anaerobes and aerobes and we set the threshold for a positive prediction for both models to a confidence level of 50%. As anticipated, there was a larger group of 2,424 strains that were predicted as negative in both models.

Next, the ability of the two models to discriminate the other states of the trait oxygen requirement was evaluated. Based on the little overlap between isopleths for true anaerobes on one hand and those for facultatively anaerobic, and microaerophilic or aerotolerant strains on the other hand in the 2D kernel density plot, the ANAEROBE model was able to distinguish most anaerobes from bacteria in all the other classes. The AEROBE model identified all true aerobes with high confidence and excluded the microaerophilic and aerotolerant strains (orange isopleths and histograms in Fig. 2B), but it failed in many cases to exclude facultative anaerobic strains. This result is reasonable since facultative anaerobes can grow without oxygen but use it for growth if present, and therefore are expected to harbor most of the protein families involved in oxygen metabolism that are also present in aerobes.

Indeed, a subsequent analysis of the feature (i.e. Pfam) importances that underlie the decision of the respective model demonstrated that the models were consistent with biochemical characteristics of the strains (Table 1). Remarkably, the predictions of the ANAEROBE model were mostly based on the absence of protein families while the AEROBE model was mostly based on their presence. The highest impact on the ANAEROBE model was determined for the absence of protein families of the oxidative decarboxylation (PF00198, PF16870, PF00676, PF00115), and oxygenases (PF01494, PF00296), as well as for the presence of protein domains related to oxidative stress tolerance (PF02915). Interestingly, the highest Gini importance was determined for the presence of the protein family of Prismane/CO dehydrogenases, whose biological role has not yet been fully understood (PF03063[33]). Our results thus point to a role of this protein family in obligately anaerobic prokaryotes. By comparison, the AEROBE model was based on the presence of proteins from the aerobic respiration, e.g. terminal oxidases (PF00115). Overall, these findings are mostly in line with established biochemistry and emphasize the capability of the models to predict oxygen requirements based on the gene inventory of prokaryotic strains. Accordingly, aerobes and facultative anaerobes also have the highest overlap in the BacDive database (intersection size, 576 strains; Fig. 2A, red arrowhead).

## Detecting distinct trait states in seemingly continuous data and determination of the underlying features

Continuous data pose a particular challenge since the boundaries between states and hence different classes are not easily defined. We chose the prokaryotic growth temperatures reported in the literature and accessible through BacDive as an example to address this problem (Fig. 3A). Certain prokaryotic strains can have a wide range of growth temperatures. As an example, *Caenimicrobium hargitense* DSM 29806 can grow between 4 and 65 °C, but is still considered a mesophile as its optimal growth temperature is about 28 °C. Therefore, temperature ranges were not considered, reducing the training data set to 75.5% of strains with temperature data where a single growth temperature was available in BacDive. However, these data points may not be the optimal growth temperature, as often only one temperature is tested or reported in the literature.

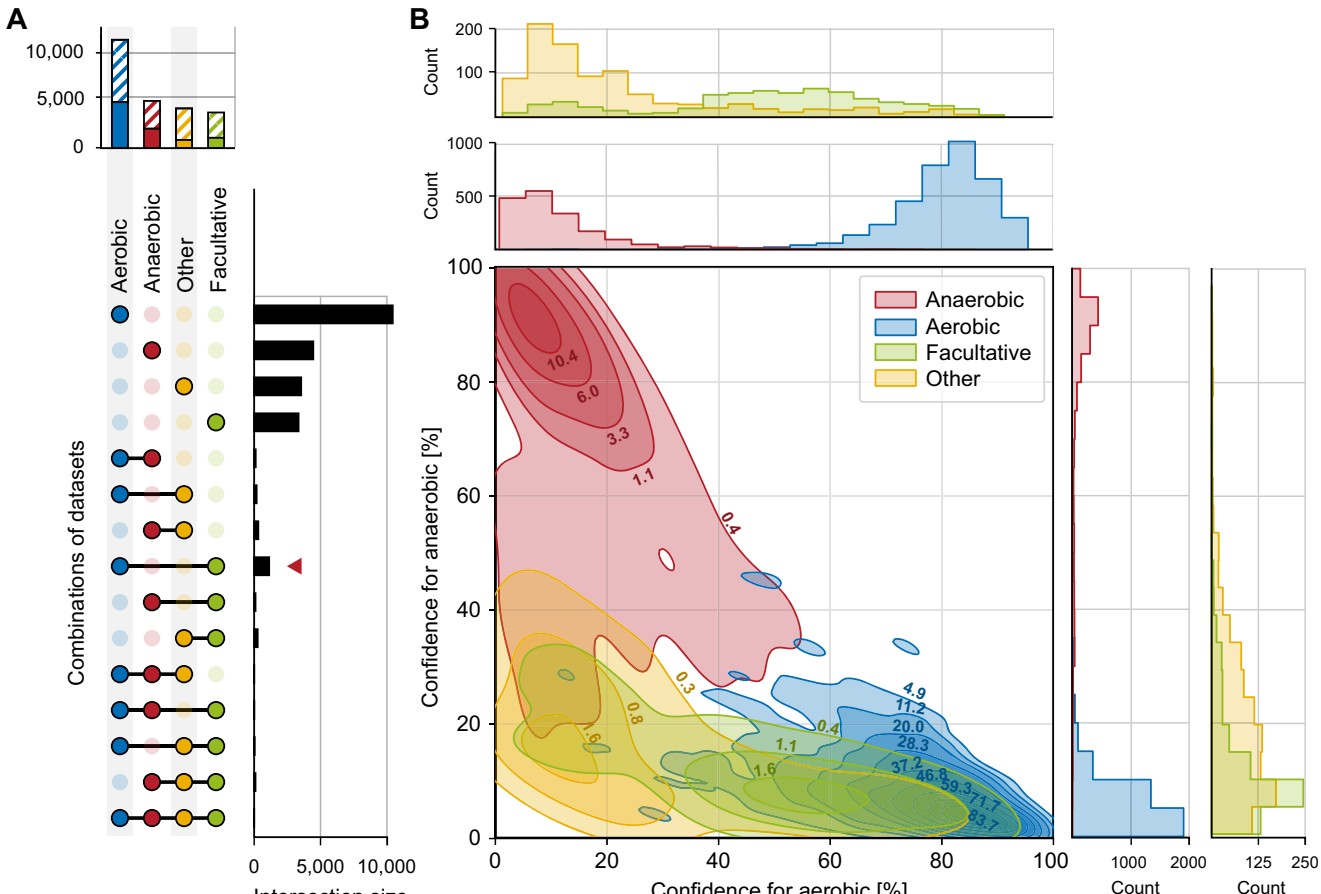

**Fig. 2 | The data basis and outcome of models for predicting oxygen requirements. A** Overview on availability and overlap of trait states for oxygen requirements of strains in the Bac*Dive* database. Obligate aerobes and obligate anaerobes were grouped with aerobes (blue) and anaerobes (red), respectively. Since aerotolerant and microaerotolerant strains had only 26 data points, they were grouped together with microaerophiles in the category 'other´ (yellow). Facultative aerobes and facultative anaerobes were also grouped together (green). The chart on top is a summary of all data available for one state of the trait. Hatched areas represent strains that had no genome sequences available and could not be used for predictions. On the right-hand side, the sizes of overlapping datasets are shown. The red arrowhead indicates the highest amount of overlapping data for aerobic and facultative anaerobic labels. **B** 2D kernel density plot and histograms of the distribution of confidence values across the full dataset. In the 2D kernel density plot, confidence values of the AEROBE model of single strains are shown on the x-axis while the confidence values of the ANAEROBE model are shown on the y-axis. The isopleths delineate all combinations of the two confidence values that were determined for the same count of strains. Isopleths are labeled with the density of data points that lie within the respective area (points per percent square). Colors represent the true state of the respective trait in these strains. Red, anaerobic; blue aerobic; green, facultative; yellow, 'other´ (microaerophilic and aerotolerant).

Bacteria are classified as thermophiles when exhibiting optimum growth temperatures of 45 °C or higher[34,35]. To establish a model (THERMO) that reliably differentiates between thermophiles and non-thermophiles and also allows to identify the underlying distribution of protein families, various limits for the classification of the two different states were tested (Fig. 3B). Our approach accounted for the fact that, in some cases, different publications report different growth temperatures for the same strain, which potentially could result in a classification of the same strain in different classes. To improve the resolution between the two classes in the THERMO model, data in a 5 °C gap at the boundary between the two states were excluded from the training. This latter strategy increased the $F_1$ score by about 10% and hence the performance of the model.

Applying the 35–40 °C boundary ($35 \leq x < 40$, with $x$ being removed) in the modeling experiment returned the lowest value for all quality metrics, most likely because it had the lowest number of data points in the training dataset, from which numerous strains that grow at 37 °C had to be excluded (Fig. 3A). However, the model employing the 40–45 °C interval as boundary condition performed well; both precision and recall were particularly high when boundaries were set to higher temperatures (Fig. 3B). Shifting the classification boundary to higher temperatures, the precision first dropped and then increased markedly reaching a peak at 60–65 °C boundary

temperature interval. However, the training dataset became increasingly unbalanced towards higher boundary temperature intervals. Since the number of data points in the training dataset was also significantly reduced at higher temperatures, the high values of the quality metrics are likely caused by overfitting.

The final THERMO model was built employing the 40–45 °C boundary and provided evidence of a genetic basis of the established classification of thermophiles (Table 2). Most evident was an increased relevance of protein families related to ribosome quality control (PF09382, PF05833), nucleotide repair (PF01612, PF03352), and for the metabolism of the compatible solutes spermidine and spermine (PF01564, PF02675). These results prove that the threshold temperature established decades ago to distinguish thermophilic from mesophilic prokaryotes has a specific biological basis and is not just a simple, man-made convention, further supported by other studies. Smith et al. identified the mesophile-thermophile temperature threshold more precisely and objectively[35] and Zheng and Wu could identify misclassified thermophilic or mesophilic bacteria based on genomic information[36].

In the future, the model performance could be increased further by training it only with optimum temperature values. Currently, too few data are available for optimum growth temperature; therefore 85% of the present

**Table 1 | Most important features used in the AEROBE and ANAEROBE models**

| Pfam | Description | AEROBE | ANAEROBE |
|------|-------------|--------|----------|
| PF03063 | Prismane/CO dehydrogenase family | −0.004 | 0.037 |
| PF00762 | Ferrochelatase | 0.017 | −0.023 |
| PF13597 | Anaerobic ribonucleoside-triphosphate reductase | −0.024 | 0.012 |
| PF01494 | FAD binding domain | 0.013 | −0.021 |
| PF00676 | Dehydrogenase E1 component | 0.012 | −0.022 |
| PF16870 | 2-oxoglutarate dehydrogenase C-terminal | 0.020 | −0.014 |
| PF00296 | Luciferase-like monooxygenase | 0.013 | −0.021 |
| PF02915 | Rubrerythrin | 0.000 | 0.031 |
| PF00487 | Fatty acid desaturase | 0.021 | −0.009 |
| PF00115 | Cytochrome C and Quinol oxidase polypeptide I | 0.015 | −0.014 |
| PF01619 | Proline dehydrogenase | 0.015 | −0.011 |
| PF00198 | 2-oxoacid dehydrogenases acyltransferase (catalytic domain) | 0.006 | −0.019 |
| PF04264 | Ycel-like domain | 0.010 | −0.013 |
| PF02391 | MoaE protein | 0.013 | −0.010 |
| PF02222 | ATP-grasp domain | 0.002 | −0.013 |
| PF00268 | Ribonucleotide reductase, small chain | 0.000 | −0.014 |
| PF17773 | UPF0176 acylphosphatase like domain | 0.000 | −0.013 |
| PF01019 | Gamma-glutamyltranspeptidase | 0.012 | 0.003 |
| PF01274 | Malate synthase | 0.012 | 0.004 |
| PF02628 | Cytochrome oxidase assembly protein | 0.011 | 0.006 |

Shown is the Gini importance that has been multiplied with −1 if the absence of a feature led to positive classification. Entries are sorted starting with the highest values for the sum of both GINI values on top.

BacDive dataset would have to be excluded from the model which in turn decreases the model performance substantially (Supplementary Fig. S2). Our findings emphasize the high demand of more standardized phenotypic information on prokaryotes for future machine learning and other applications of artificial intelligence.

## Improving models through iterations of machine learning and expert curation

In contrast to oxygen requirement and growth temperature, motility as such is a binary trait since a bacterium can be either motile or not. However, motility is conferred by two types of mechanisms, flagellar motion and gliding. The first type can include several different flagellum arrangements and motor proteins. The latter does not depend on flagella for movement but instead on membrane proteins, type IV pili or polysaccharide jets. Existing models of previous studies diverged in terms of their capabilities for predicting motility. Whereas a successful model reached a higher accuracy (the only available metric) of 0.93[15] other models returned only accuracies of 0.83 to 0.88 which are unsatisfactory[13,14,37]. A closer inspection of the underlying modeling procedures revealed that the superior models had been trained solely on motion conferred by flagella while the poor models did not distinguish between the known types of motilities. When limited to data sets with genome sequences, BacDive offers a total of 7090 standardized datapoints for the binary trait 'motility' with 2899 motile and 4191 non-motile strains, but the type of motility (flagella or gliding) is only known for less

than 16.0% (464) of the motile strains (Fig. 4A). Among the latter, 68.7% (319) were known to exhibit flagellar motion that could be further classified based on the arrangement of the flagella. To test the impact of the mode of motility on the model quality, we started to build models with the whole motility dataset and then iteratively engineered the input of data.

The initial MOTILE_0 model was trained using data for all strains from the BacDive database for which both, annotation of motility and genome sequences were available. The 7090 strains in the dataset had an acceptable distribution of 1:1.4 between the two classes 'motile' and 'non-motile' (Fig. 4B). As anticipated based on previous modeling attempts in the literature with just two classes[13–15,37], the model performed mediocre with an AUPR of 0.87 (Fig. 4C) and an F1 score of 0.83 (Fig. 4B), a recall of 80.9% and a precision of 84.2%. A closer look into the feature importances of this first model revealed that decisions were based on the presence of flagellar proteins (Supplementary Table S1). In line with this observation, 89% of the false predictions for which the type of motility was known were annotated as gliding. Hence, this simple model of binary trait states fails to predict the motility which is based on a gliding mechanism.

In the next iteration, the input datasets were engineered to improve the model. First, strains known to exhibit gliding motility were omitted from the input datasets which resulted in a small increase of both, the AUPR and the $F_1$ score to 0.88 and 0.86, respectively (MOTILE_1 model, Fig. 4C, recall: 82.0%, precision: 86.5%). Since the group of motile strains with an unknown type of motility is likely to contain a larger fraction of gliding bacteria, we limited the positive group to those strains with proven flagellar motility in our next iteration (MOTILE_2 model). This iteration is comparable to one of the modeling attempts of the past[15]. An in-depth analysis showed that protein families with highest importance for the predictions of the MOTILE_2 model were all part of the protein complex forming the flagellum (Table 4), except for one that is part of the Type III secretion system (PF18269), confirming that the model is based on the molecular inventory known for flagellar motility. Surprisingly, however, the performance of this model dropped to F1 score of 0.70 (Fig. 4) and precision and recall decreased markedly to 68.9% and 70.6%, respectively. While the imbalance of the dataset led to an increase in the AUC to 0.95, the AUPR lowered to 0.74, reflecting the performance drop mentioned above and further supporting the choice for AUPR over AUC. A comparison of the different classes revealed that a large number of the protein families that determined the predictions were not only limited to flagellated bacteria but also occurred in up to 26.5% of the non-motile strains, explaining the large number of false positives. The same is also true for the MOTILE_0 and MOTILE_1, but these false positives were diluted by the larger datasets in these first modeling iterations.

This seeming contradiction prompted us to analyze the actual distribution patterns of the protein families associated with flagellar motility (Table 3) across the genome sequences of non-motile strains. As expected, the largest part of these genomes (>75%) encoded none of the protein families. However, 557 strains (13.4%) had more than 10 of the flagellum protein domains, and of these 557 strains, 327 had even ≥ 19 of the queried 21 protein families (Fig. 5A).

In the next step, we removed the 327 strains that encoded large parts of the protein families needed to synthesize a flagellum from the data set (≥19 Pfams annotated, as highlighted by red shading in Fig. 5A). This improved the performance of the resulting MOTILE_2+ model significantly compared to the MOTILE_2 model, reaching an $F_1$ score of 0.95 (Fig. 4) and a precision of 93.8%. The iterations of the MOTILE model clearly exemplify how machine learning models can be improved significantly when several iterations of modeling are conducted, and these alternate with assessments of the previous model output based on the knowledge of the underlying biological mechanisms.

The 327 strains that were found to contain the majority of protein families (Pfam) associated with flagellar motility represent interesting targets for future detailed growth experiments because motility genes might not be expressed and hence not been detectable under the specific

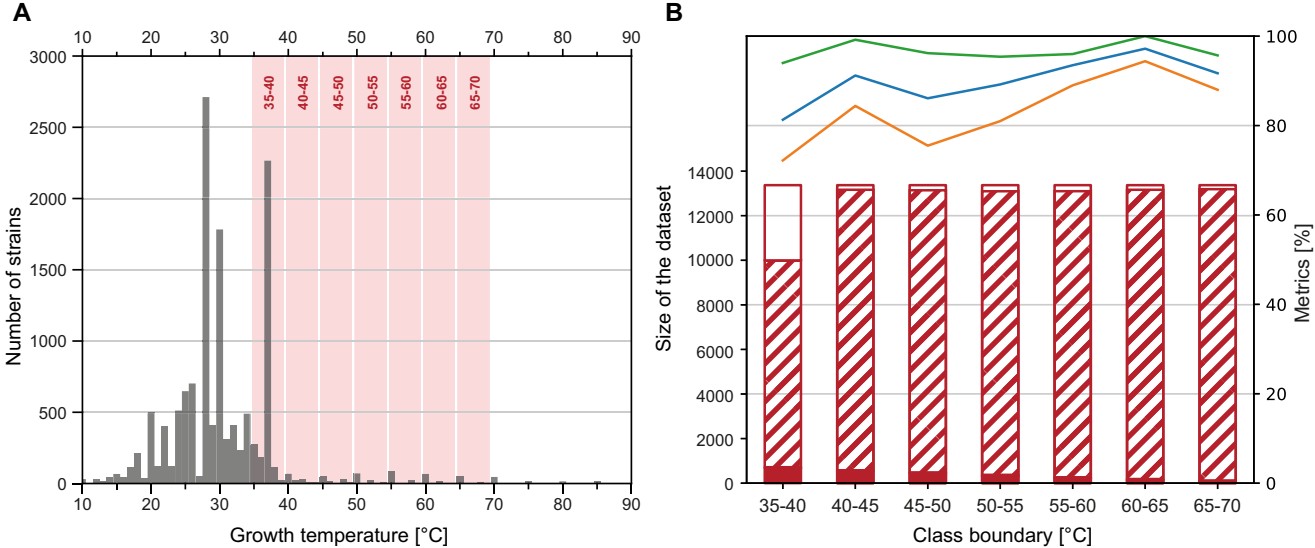

**Fig. 3 | Supporting data, input data, and performance of the THERMO model for prokaryotic growth temperature. A** Overview on growth temperature data available in the BacDive database, limited to strains with available genome sequences. Shown is the number of strains with reported growth in a specific temperature range of 1 °C. The temperature ranges that are excluded to delineate the datasets for the model runs depicted in B are shown in red. **B** Evaluation metrics of the THERMO model, trained and tested with different subsets of the data. The x-axis gives the specific 5 °C-interval applied to delineate the two classes in the individual model run. The y-axis for the column chart is on the left. Hatched areas of columns mark the amount of truly negative data, filled areas the amount of truly positive data, empty areas the amount of discarded data from within the 5 °C-interval. The y-axis for the line chart is on the right. Green, recall; orange, precision; blue, $F_1$ score.

cultivation conditions employed in previous laboratory studies. For instance, certain bacteria (e.g. *Actinoplanes*) are immotile in their vegetative stage and only form flagellated spores. Of the strains with 21 protein families for flagella genes, a sample of 5 was selected to test for possible mobility. Four of them showed mobility in soft agar (Supplementary Table S2), one of which moved particularly far in the experiment (Fig. 5B, DSM 9356). This experiment serves as a proof-of-concept to demonstrate how a machine learning model can also identify phenotypes that are falsely reported in the literature.

### Model performance across distinct phyla

Our evaluation of model performance across different bacterial and archaeal phyla demonstrates high overall accuracy, with a median classification accuracy of 94.9% and a median prediction confidence of 91.6% (Fig. 6). This generally robust performance highlights the suitability of our Random Forest-based approach for predicting phenotypic traits from genomic annotations across diverse microbial lineages. However, caution has to be exercised when interpreting results for phyla with low representation, particularly the 11 phyla with fewer than 10 annotated traits each, as the limited sample size renders their accuracy and confidence metrics statistically uncertain.

Well-studied phyla with a high sample representation such as *Bacillota*, *Pseudomonadota*, and *Actinomycetota* consistently yielded high classification accuracy and prediction confidence, underscoring the effectiveness of our models when sufficient genomic and phenotypic data are available. Interestingly, certain less represented phyla, such as *Mycoplasmatota* and *Thermodesulfobacteriota*, also showed excellent prediction accuracy, indicating that the genomic markers were tightly associated with phenotypic traits, conserved and characteristic for these groups. In comparison, archaeal phyla, although taxonomically distinct from bacteria, displayed notably lower median prediction confidence despite moderate accuracy of about 95%, except for *Nitrososphaerota*, whose extremely limited sample size (*n* = 6) further complicates a reliable interpretation. The reduced confidence in archaeal predictions might be caused by unique genomic features and differences in annotations as compared to bacterial reference genomes, emphasizing the need for further sequencing and annotation refinement within the archaeal domain.

Notably, certain bacterial phyla yielded unexpectedly low performance metrics. *Calditrichota*, which were only represented by one strain with 3 annotated traits, displayed both very low classification accuracy and prediction confidence, which may reflect unique and/or divergent genomic signatures of this particular strain that are poorly captured by the current feature representation, or significant annotation gaps. Despite acceptable classification accuracies (~90%) and coverage by 83 and 34 species, respectively, *Deinococcota* and *Chloroflexota* also exhibited median prediction confidences below 80%, which can be attributed to an increased genomic variability within these taxa. Both phyla contain species highly resistant to extreme environmental conditions, and are known extremophiles. These adaptations are reflected in unique molecular signatures, such as conserved signature indels (CSIs) and proteins (CSPs) in *Deinococcota*[38]. Members of *Chloroflexota* are known to comprise a diversity of phenotypes, including anoxygenic phototrophs, aerobic thermophiles and the use of unusual or even toxic compounds as electron acceptors[39]. This diversity makes protein-based prediction challenging. However, since classification accuracies are still acceptable in both cases, we assume high reliability of the models even at the lower confidence values.

Our findings suggest that phylogenetic representation alone is not the primary determinant of model robustness. Instead, the extent to which a phylum is metabolically or ecologically distinct appears to have a greater effect on predictive performance. Taxa adapted to specialized or extreme niches—such as *Deinococcota* and *Chloroflexota*—pose particular challenges due to their unique genomic features and divergent protein repertoires, which are often underrepresented in standard annotation databases. Thus, beyond increasing taxonomic coverage, improving model generalizability will require incorporating a broader spectrum of functional and environmental diversity into training datasets.

### Exploiting high quality predictions can significantly increase the body of knowledge in databases

For the predictions of the different models to augment the phenotypic data in databases such as BacDive, the quality of the predictions needs to be evaluated. Accuracy is an often-employed metric for evaluating the performance of models[14,15,37]. However, accuracy is not a good choice as soon as

**Table 2 | Feature importances of the THERMO model with a boundary of more than 45 °C for positive and less than 40 °C for negative classification**

| Pfam | Description | Importance | Function |
|---|---|---|---|
| PF16658 | Class II release factor RF3, C-terminal domain | −0.019 | Translation termination |
| PF09382 | RQC domain | 0.017 | Ribosome quality control |
| PF01867 | CRISPR associated protein Cas1 | 0.013 | Prokaryotic defense |
| PF05833 | NFACT N-terminal and middle domains | 0.013 | Ribosome quality control |
| PF03352 | Methyladenine glycosylase | −0.011 | DNA repair |
| PF02675 | S-adenosylmethionine decarboxylase | 0.010 | Biosynthesis of spermidine and spermine |
| PF03976 | Polyphosphate kinase 2 (PPK2) | −0.009 | PolyP metabolism |
| PF01564 | Spermine/spermidine synthase domain | 0.009 | Biosynthesis of spermidine and spermine |
| PF01612 | 3'-5' exonuclease | −0.008 | DNA repair |
| PF00255 | Glutathione peroxidase | −0.008 | Protection against hydroperoxides |

Shown is the Gini importance that has been multiplied with −1 if the absence of a feature led to positive classification.

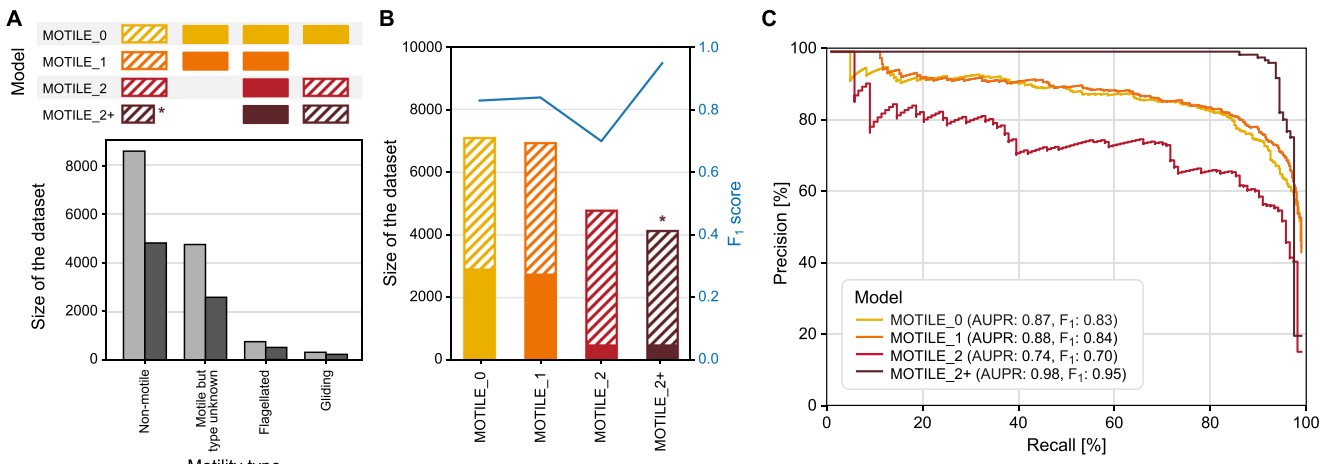

**Fig. 4 | Data availability and performance of the different models for the prediction of prokaryotic motility. A** Availability of data points for different types of motility in the Bac*Dive* database. Light grey columns give the total number of data in the Bac*Dive* database and dark grey columns the number of strains for which of these strains genome sequences were available. The information above indicates which datasets have been used as positive (filled) or negative (hatched) labels for the different models. **B** Size of the datasets available for the four MOTILE models (columns) and corresponding F1 scores (blue line). Hatched bars give numbers of negative, filled bars those of positive data (same as in panel A). The corresponding datasets for the different motility mechanisms are depicted in panel A. The dataset marked by * was reduced by strains that did contain more than 10 flagellum proteins. **C** Precision-recall curves showing the performance of the four different versions of the MOTILE model.

the input data are imbalanced which is often the case for biological datasets. To evaluate this further, we also included a dataset that would otherwise be inappropriate for the application of machine learning because of its high imbalance in the training dataset. The ACIDO model should predict whether a strain can grow at a pH below 5 or not. As mentioned, the particular input dataset was highly imbalanced with 2483 strains being in the negative but only 45 strains being in the positive class. Most of the strains were indeed correctly predicted as negative and the ACIDO model had an accuracy of 97.3%, falsely suggesting a very good performance. After all, the model had a precision of only 13% and a recall of 75%, emphasizing that this metric in inappropriate for such kind of data. Therefore, we evaluated a number of metrics established (see Materials and Methods) for all eight different machine learning models developed in the present study to assess the quality of their predictions (Table 4, Supplementary Fig. S3A).

Besides the ANAEROBE, AEROBE, THERMO, MOTILE2 +, ACIDO models, we also built three additional models for which sufficiently large datasets are available through the Bac*Dive* database (Fig. 4). The model GRAMPOS was used to predict Gram-positive bacteria, SPORES to predict the capability of a strain to form spores and PSYCHRO was established to predict psychrophily (i.e., a growth optimum below 20 °C).

As stated, accuracy is a poor choice to evaluate the performance of models as soon as data are unbalanced, which is demonstrated by the fact that all models have an accuracy of more than 90% (Table 6). Although ROC curves are a good means to visualize the performance, the AUC shares similar weakness as accuracy. The area under the precision/recall curve (AUPC) has proven to be a better visualization for unbalanced datasets[40]. Precision, recall, specificity, and NPV are metrics that summarize parts of the confusion matrix and cover different aspects of the quality of a model. These metrics can be relevant to assess specific aspects, e.g. if false positives must be avoided, a high precision is needed, and the recall is less important. However, none of those metrics can reflect the model quality by itself, e.g. the PSYCHRO model has a high specificity of 97.7%, but a drastically low precision of 6.2%. Thus, more complex scores that take several aspects of the confusion matrix into account were evaluated. Although the *norm*MCC is the only metric that takes all four rates of the confusion matrix into account, we selected the $F_1$ score because it is easier to interpret and to compare between models while being almost as sensitive to imbalanced data as the *norm*MCC metric. However, it should be mentioned that the true negative samples are not considered for the calculation of F1 scores which is shown by the AEROBE model that had higher F1 scores in comparison to

**Table 3 | Feature importances of the MOTILE2 model**

| Pfam | Description | Importance | in Pos (%) | in Neg (%) |
|---|---|---|---|---|
| PF02561 | Flagellar protein FliS | 0.019 | 84.03 | 17.15 |
| PF14842 | FliG N-terminal domain | 0.018 | 98.28 | 25.37 |
| PF14841 | FliG middle domain | 0.017 | 98.28 | 25.42 |
| PF02154 | Flagellar motor switch protein FliM | 0.015 | 92.87 | 19.17 |
| PF00700 | Bacterial flagellin C-terminal helical region | 0.014 | 98.53 | 25.42 |
| PF02049 | Flagellar hook-basal body complex protein FliE | 0.014 | 98.77 | 24.85 |
| PF01706 | FliG C-terminal domain | 0.012 | 98.77 | 25.73 |
| PF00669 | Bacterial flagellin N-terminal helical region | 0.011 | 98.53 | 25.29 |
| PF07195 | Flagellar hook-associated protein 2 C-terminus | 0.010 | 85.01 | 17.28 |
| PF01313 | Bacterial export proteins, family 3 | 0.010 | 98.53 | 25.77 |
| PF18269 | T3SS EscN ATPase C-terminal domain | 0.010 | 98.03 | 25.51 |
| PF01052 | Type III flagellar switch regulator (C-ring) FliN C-term | 0.010 | 99.02 | 26.52 |
| PF00813 | FliP family | 0.008 | 98.03 | 25.86 |
| PF06429 | Flagellar basal body rod FlgEFG protein C-terminal | 0.006 | 99.02 | 25.59 |

Shown is the Gini importance.
All features had positive influence. Pos = feature found in % of the positive group, Neg = feature found in % of the negative group.

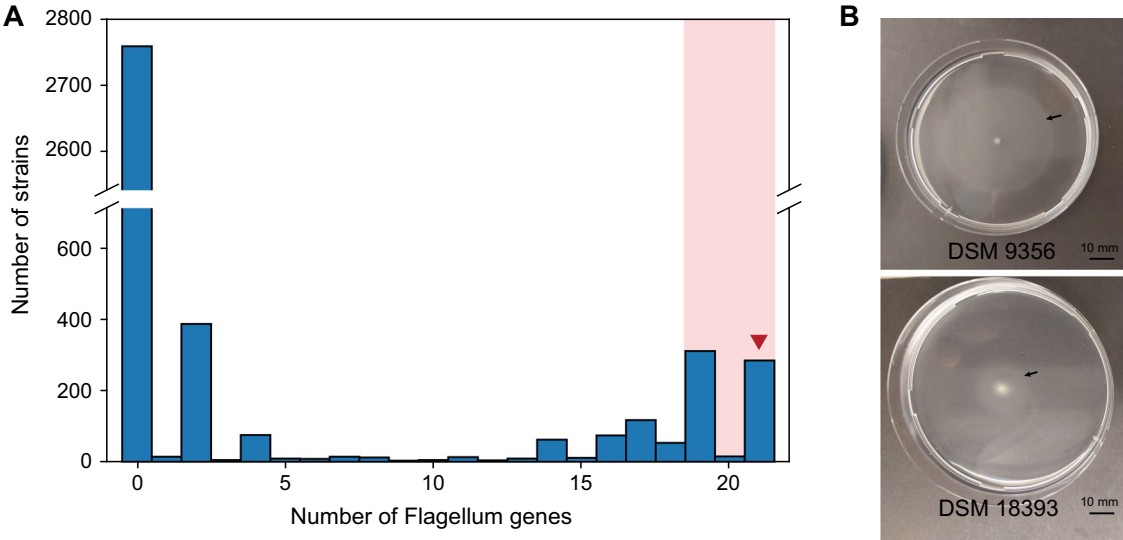

**Fig. 5 | Supporting data for the MOTILE_2+ model. A** Number of flagellum genes in strains that were reported as non-motile in the literature. Shaded in light red are strains that were excluded from the negative training set in the MOTILE_2+ model. The red arrowhead indicates groups of strains of which representatives were tested for motility in laboratory experiments. **B** Experimental proof of motility of two bacterial strains on semi-solid agar. The arrows point to the edge of the opaque area to which the cells have moved. Both have been reported as non-motile in the literature.

*norm*MCC, since the latter takes also the negative predictive value (NPV) into account. A cutoff of 0.8 was applied to the $F_1$ score of a model, thus, the highly imbalanced and underperforming models PSYCHRO and ACIDO were not used to generate predictions from prokaryotic genomes and integrate them into the Bac*Dive* database.

We then applied the six selected models to all 15,938 genomes in the dataset. This generated a new dataset with 95,628 data points. Of these, 45,232 overlapped with experimental data existing in the database and thus can be used to check for possible errors in the experimental results. The remaining 50,396 data points constitute completely new information that was lacking so far and thus complement the existing 1,247,971 manually curated phenotypic data in the Bac*Dive* database. For the selected traits, this represents an increase in data availability of more than 160%. It is also worth mentioning that more than 2000 strains had no single datapoint for the predicted phenotypic traits before and were thus significantly enriched in

strain-associated information (Supplementary Fig. S4). In summary, 53% (50,396) of the predicted data were newly generated information, 45% (42,624) of the predictions were for already manually curated data in Bac*Dive* and agreed with them, whereas only less than 3% (2608) of all predicted data were for manually curated data that contradicted the existing information (Supplementary Fig. S4B). In only 113 cases, predictions contradicted the existing Bac*Dive* data with over 90% confidence. In 27 cases, the predictions identified actual errors in the Bac*Dive* annotation, allowing for corrections. 32 cases related to oxygen tolerance predictions where literature listed both anaerobic and aerobic growth, demonstrating that, in fact, predictions and original Bac*Dive* entries differed but were both correct.

Because of the high quality of the predictions obtained by the six machine learning models established, we included the predicted data alongside the manually curated data in the Bac*Dive* web interface (Fig. 7, left). Additionally, we show an overview on all predicted data for the

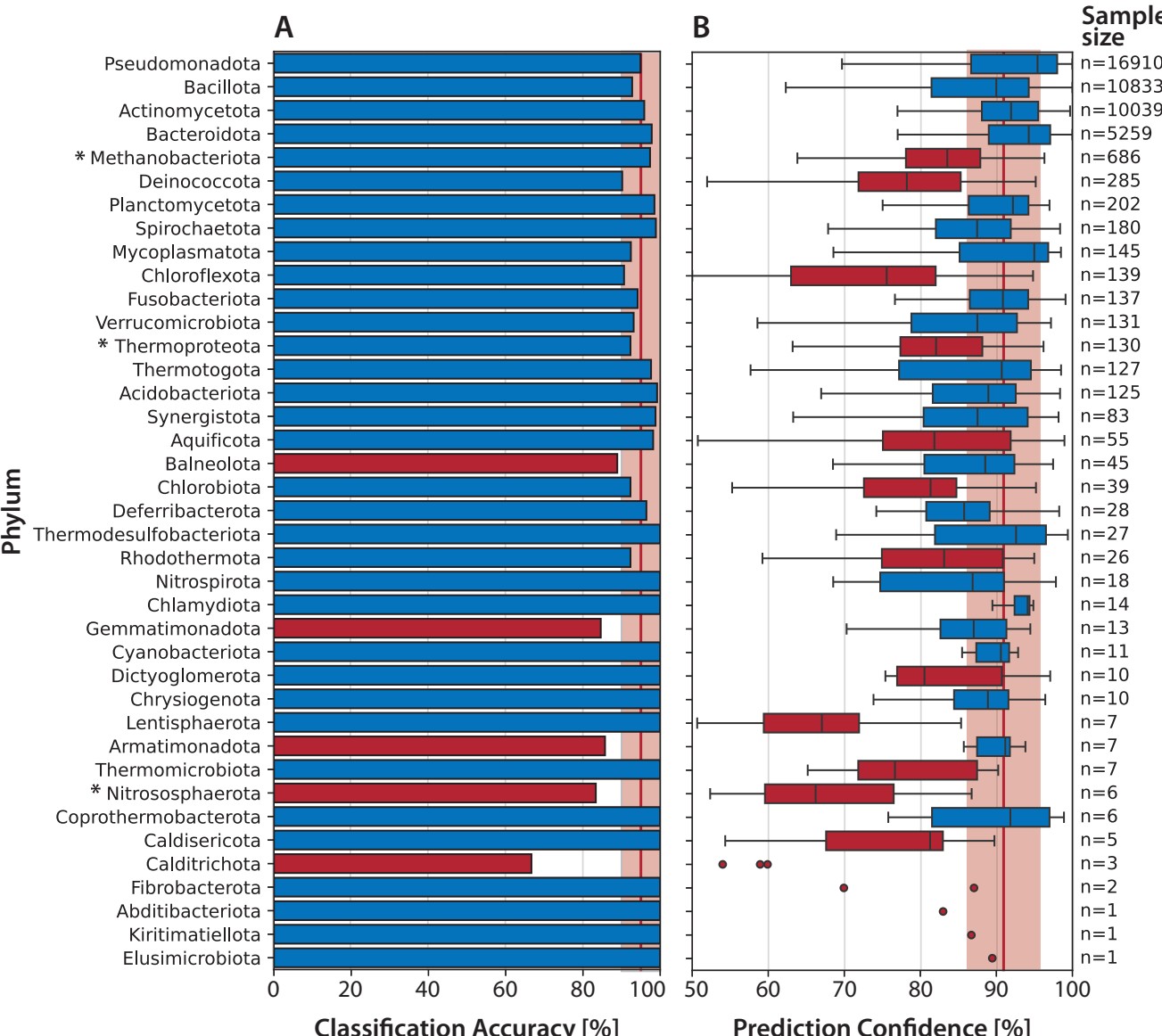

**Fig. 6 | Taxonomic variation in model performance across different bacterial and archaeal phyla. A** Classification accuracy (percentage of correctly predicted phenotypic traits) per phylum. **B** Distribution of prediction confidences (probability scores assigned by Random Forest models) across phyla. Phyla belonging to the archaeal domain are marked with an asterisk (*). The vertical red lines indicate the overall median across all phyla, with a surrounding 5% interval shown as a shaded red area. Phyla with a classification accuracy or median prediction confidence below the 5th percentile are highlighted in red, indicating reduced performance in these groups. Sample sizes for each phylum are shown on the right. Please note that these numbers represent the number of annotated traits per phylum, not the number of strains.

organism, as well as the information from which genome the data has been derived (Fig. 7, right).

**Perspectives and limitations**

The results of our analysis highlight the importance of the quantity and quality of balanced training data for generating trustable predictions for phenotypic traits of prokaryotes from their genome sequences. Of similar importance is the consideration of available biological knowledge. As demonstrated in this work, generalized models easily fail to predict the actual phenotype, if well-known differences in the underlying biology are not taken into account during the design of the model. Our approach makes it possible to explicitly determine the key underlying genetic features and hence offers the opportunity for a deeper biological interpretation of the results. Not all commonly used metrics are equally suitable to evaluate machine models, though. Metrics that integrate several quality aspects, in particular *norm*MCC or the F1 score should preferably be applied to assess the quality of AI predictions. If these conditions are met, highly performing machine learning models generate valuable novel information that can be used to enrich existing databases. In our exemplary study, six models yielded 50,396 completely new datapoints for 15,938 strains and allowed to challenge 45,232 existing datapoint yielding consensus in more than 97% of the cases. Thus, high quality, machine learning models that take biological insights into account clearly provide the means to significantly improve the availability of data in open databases for subsequent functional studies.

To enhance the predictive performance and biological relevance of machine learning approaches, a systematic expansion of the presently available phenotypic and genomic datasets remains crucial. Our analyses highlight that the prediction of phenotypic traits by machine learning could be improved significantly if the quality and quantity of genomic data were increased (Fig. 1). Targeted sequencing efforts towards strains from microbial culture collections that have already been phenotypically characterized, would be particularly promising.

**Table 4 | Model performance for the independent test dataset**

| Model | Accuracy | Precision | Recall | Specificity | NPV | F₁-score | AUC | AUPC | normMCC |
|---|---|---|---|---|---|---|---|---|---|
| GRAMPOS[a] | 96.3% | 98.1% | 92.5% | 98.8% | 95.3% | 0.95 | 0.99 | 0.97 | 0.92 |
| SPORES[a] | 95.6% | 92.9% | 93.6% | 96.6% | 96.9% | 0.93 | 0.98 | 0.94 | 0.90 |
| ANAEROBE[a] | 97.8% | 90.3% | 99.1% | 97.5% | 99.8% | 0.94 | 0.99 | 0.98 | 0.93 |
| AEROBE[a] | 91.1% | 94.3% | 89.9% | 92.7% | 87.3% | 0.92 | 0.96 | 0.96 | 0.82 |
| MOTILE2+[a] | 98.1% | 93.8% | 96.0% | 98.6% | 99.1% | 0.95 | 0.99 | 0.98 | 0.94 |
| THERMO[a] | 98.9% | 74.7% | 99.2% | 98.9% | 100.0% | 0.85 | 0.99 | 0.94 | 0.86 |
| PSYCHRO | 97.7% | 6.2% | 85.7% | 97.7% | 100.0% | 0.12 | 0.89 | 0.31 | 0.23 |
| ACIDO | 97.3% | 13.0% | 75.0% | 97.4% | 99.9% | 0.22 | 0.82 | 0.46 | 0.31 |

The data is shown in percent (multiplied with 100) to improve readability.

*AUC* area under the curve, *AUPR* area under precision/recall curve, *norm*MCC = normalized Matthews correlation coefficient, *NPV* negative predictive value.

[a]The high-quality predictions of these models were integrated into the Bac*Dive* database and marked as synthetic data (see text).

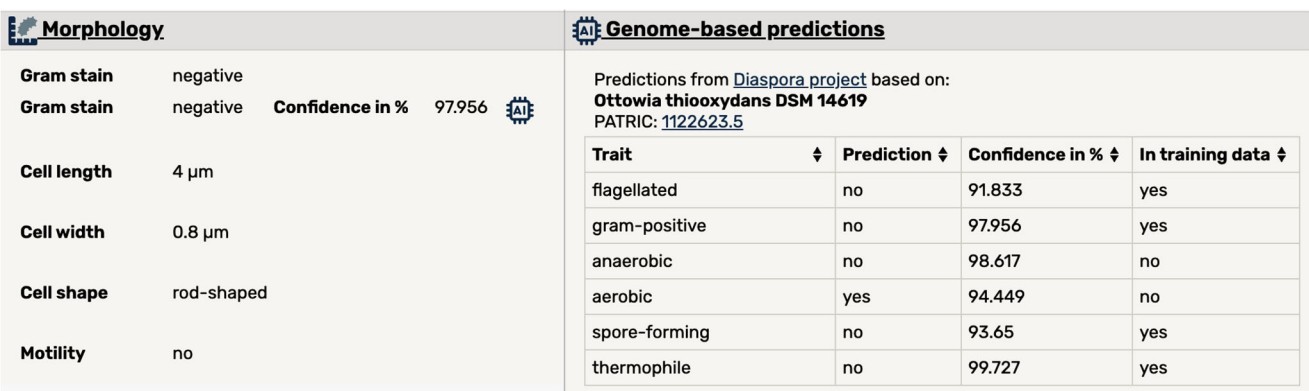

**Fig. 7 | Screenshot of the web interface showing predictions in the Bac*Dive* database.** Left: Predicted data are integrated alongside manually annotated data. The shown confidence value and an AI icon indicate that the dataset was derived computationally. Right: Table gives an overview on the predictions currently available for a strain, as well as the genome from which the prediction has been derived.

In parallel, future research should focus on the experimental validation of computational predictions, such as laboratory assays on genomically characterized but phenotypically unannotated strains. Ultimately, applying these models to metagenome-assembled genomes (MAGs) would extend predictions towards growth conditions of even uncultivated microbes. This would also require deducing growth medium composition needed for cultivation. In the context of the Media*Dive* database[41], we are currently exploring complementary approaches, aiming at the prediction of nutrient and chemical requirements of uncultured microorganisms. Such integrated strategies would considerably strengthen both the reliability and applicability of predictive phenotype modeling in microbiology.

## Methods
### Experimental design
The objective of this study was to develop and validate machine learning models for predicting phenotypic traits of prokaryotes from genomic data, utilizing the comprehensive Bac*Dive* database. We selected traits with high data availability, focusing on those with more than 3000 strains, and addressed both binary and multi-state classifications, such as Gram-staining response and oxygen requirements. Random Forest algorithms were employed to build robust models, which were trained and validated using cross-validation techniques. Feature importance analysis provided biological insights into the predictors used.

### The genotype dataset
We selected 15,938 strains from Bac*Dive* for which genome sequences were available. For each species that was represented by several strains, the strain with the highest-quality genome was chosen (see below). Of the selected strains, 15,538 were *Bacteria* and 356 were *Archaea* including 12,519 type strains. A complete taxonomic overview can be found in Fig. S5. For each strain chosen, the genome sequence of highest quality was selected based on genome completeness, number of contigs, and on an exact match of the NCBI Taxonomy-ID to the corresponding Bac*Dive* strain. If only a matching on the species level was possible, i.e. when the strain is not represented in the NCBI taxonomy, a species level matching was accepted. Genome sequences were downloaded from the openly accessible databases NCBI and BV-BRC (formerly PATRIC)[42,43].

All genomes were translated into protein sequences using Prodigal[44] and annotated using the InterProScan tool v 5.63–95.0[26], including the module to detect protein families by Pfam in the dataset. For the latter, an e-value threshold of $10^{-20}$ was chosen. The presence and absence of Pfams was subsequently transformed into a vector consisting of the digits 1 and 0, respectively.

### The phenotype dataset
Phenotypic data were selected from a list of more than 1000 data fields from the Bac*Dive* database[11] based on the availability of data. In total, 80% of the data were used for the training of the model and 20% were used as test dataset (see below). Ambiguous data, in particular strains that had both, positive and negative labels, occur either when different publications contradict each other or when test results are unclear. Such ambiguous data were removed from the training dataset. The feature space was further reduced by removing features with low variance (<20%) for each of the traits.

### The machine learning algorithm

We examined different supervised-learning and deep learning algorithms in this study. Among supervised learning methods, Support Vector Machines (SVM) with the Radial Basis Function (RBF) kernel performed best, yielding $F_1$ scores (cf. next section) that were up to 5% higher than scores obtained by other techniques. However, as one major goal of our study was the biological interpretation of the model, non-linear kernels such as RBF cannot be used as the latter does not allow to determine the importance of individual features. The same is true for all deep learning methods, such as artificial neural networks (ANNs). When SVMs with a linear kernel were tested, they were found to perform worse or at best similar to Random Forest algorithm.

While advanced deep learning algorithms such as neural networks can achieve higher predictive accuracy, our priority was the (biological) interpretation of the results to gain insights into the genetic basis of the traits investigated. Especially deep learning models are known to obscure underlying decision processes, complicating biological interpretation downstream[21]. Preliminary analyses showed that performance gains from these methods were limited relative to the interpretive clarity provided by Random Forests. Therefore, we employed Random Forest as a pragmatic balance between predictive accuracy, computational feasibility, and feature interpretability. We like to emphasize at this point that other machine learning algorithms would have been suitable as well, e.g. gradient boosting or support vector machines.

The random forest models were trained using consistent hyperparameter settings across all traits, specifically 200 decision trees and a maximum tree depth of 10. These hyperparameters were determined based on preliminary grid search analyses, where multiple combinations of tree counts (50–500) and tree depths (5–20) were systematically evaluated using cross-validation. The chosen settings provided an optimal balance between predictive accuracy, computational efficiency, and model interpretability, particularly concerning reliable estimation of feature importances through multiple runs of the model. A classification threshold of 0.5 was uniformly applied to translate predicted probabilities into binary trait predictions. While alternative thresholds may slightly optimize predictions for specific traits, the 0.5 threshold was chosen as a standard reference point to ensure comparability across traits, recognizing that trait-specific threshold optimization could be beneficial in future work. All of the algorithms were implemented using the Python scikit-learn library v1.3.0.

Feature importances were derived from the classifier and were computed as the (normalized) decrease in model performance upon elimination of the individual feature (i.e., Pfam) from the model, quantified as the Gini importance (see below)[45].

### Statistics and reproducibility

All models were evaluated using stratified 10-fold cross-validation to ensure robust estimation of performance across imbalanced trait datasets. For each trait, models were trained on 80% of the data and evaluated on the remaining 20%. Performance metrics were averaged across folds. No biological replicates were used, as this study is based entirely on computational prediction using curated genomic and phenotypic datasets. The dataset includes 15,938 prokaryotic strains and over 133,000 trait annotations. All code and datasets are publicly available, ensuring full reproducibility of the analyses presented.

### Evaluation metrics

All assessed metrics are the result of 5-fold cross validation. For this purpose, the data were split into five sets of equal sizes. One of the subsets was used as test data, while the remaining four sets were used to train the model. This was then repeated using each of the five different subsets as test data and the remainder as training sets. As the model is trained and tested with different parts of the data, this method is used to recognize overfitting of the model.

All metrics for binary characters are described using a standard confusion matrix in which the actual class of a test data set is compared to the predicted class:

|  |  | Actual condition | |
|---|---|---|---|
|  |  | Positive | Negative |
| Predicted condition | Positive | True positive (TP) | False positives (FP) |
|  | Negative | False negative (FN) | True negative (TN) |

**Accuracy** is the most commonly used metric to evaluate the predictiveness of a machine learning model. It is calculated by dividing all true predictions by the total number of observations. The resulting accuracy is a value within the range from 0 to 1, with 1 being the perfect prediction, 0.5 being no better than random guessing and 0 being an absolute contradiction:

$$\text{Accuracy} = \frac{\text{TP} + \text{TN}}{\text{TP} + \text{TN} + \text{FP} + \text{FN}} \qquad (1)$$

**Precision** indicates the ratio of true positive results to all positive predictions:

$$\text{Precision} = \frac{\text{TP}}{\text{TP} + \text{FP}} \qquad (2)$$

**Recall** (also referred to as **sensitivity** or **true positive rate**) is the ability of the model to correctly predict positive samples among all actual positives:

$$\text{Recall} = \frac{\text{TP}}{\text{TP} + \text{FN}} \qquad (3)$$

**Specificity** (also referred to as **true negative rate**) is the ability of a model to correctly reject negative samples among all actual negatives:

$$\text{Specificity} = \frac{\text{TN}}{\text{TN} + \text{FP}} \qquad (4)$$

**Negative predictive value (NPV)** is the ratio of true negative results to all negative predictions:

$$\text{NPV} = \frac{\text{TN}}{\text{TN} + \text{FN}} \qquad (5)$$

**The $F_1$-score** is the harmonic mean of specificity and recall and thus represents both in one value. It should be used when both values should be as high as possible.

$$F_1 = \frac{2\text{TP}}{2\text{TP} + \text{FP} + \text{FN}} \qquad (6)$$

**The receiver operating characteristic (ROC) curve** evaluates a classifier's ability to distinguish between two classes by plotting the true positive rate (recall) against the false positive rate over varying thresholds. Typically, a threshold of 0.5 is used to classify a trait as positive, but adjusting it from 0 to 1 shifts the trade-off between true and false positives. The **area under the curve (AUC)** quantifies the model's overall discriminative power (see Fig. 4C and Supplementary Fig. S3B).

The precision-recall (PR) curve, in contrast, focuses on model performance in imbalanced datasets by plotting precision against recall. This

curve is particularly useful when positive instances are rare, as it is less affected by the number of true negatives. The **area under the PR curve (AUPR)** provides an alternative measure of predictive quality, especially in cases where precision is more critical than overall classification performance.

**The Matthews correlation coefficient (MCC)** is a widely used metric for binary classification that is particularly robust for imbalanced datasets, as it incorporates all four components of the confusion matrix (true and false positives and negatives) into a single score[46,47]. Unlike metrics such as accuracy, which can be misleading in the presence of class imbalance, MCC yields values between –1 (inverse prediction) and +1 (perfect prediction), with 0 indicating random performance. To facilitate comparability with other metrics reported on a [0, 1] scale, we linearly rescaled the MCC using min-max normalization and refer to this transformed score as *norm*MCC.

$$\text{MCC} = \frac{\text{TP} \times \text{TN} - \text{FP} \times \text{FN}}{\sqrt{(\text{TP} + \text{FP}) \times (\text{FN} + \text{TN}) \times (\text{TP} + \text{FN}) \times (\text{FP} + \text{TN})}} \quad (7)$$

$$\textit{norm}\text{MCC} = \frac{\text{MCC} + 1}{2} \quad (8)$$

**Confidence** contrasts with the previously mentioned metrics, as it is not a method to access the quality of a model but the quality of a single prediction. Thus, it is a measure of certainty given by the classifier for each sample. Thus, a confidence of 0.8 means that the model is 80% sure that a data point is positive. We applied a threshold for a positive prediction of 0.5 throughout this study. If the confidence for a sample is 0.4, the accepted prediction of the sample is negative with 60% certainty. How the confidence is calculated depends on the algorithm used. In case of random forest, the predicted class probabilities of an input sample are calculated as the mean predicted class probabilities of the trees in the forest. The class probability of an individual tree is the proportion of samples of the same class in a leaf.

The 2D kernel density plot for the analysis of the model predicting oxygen requirement (cf. Fig. 2) was generated using the seaborn Python package and the Gaussian kernel estimation employed with the scipy package. The dataset available for the oxygen requirements of strains comprised 8405 data points. The confidences (see above) for all samples from the AEROBE and ANAEROBE model were depicted on the x and y axis, respectively, and colored by their actual oxygen requirement according to the literature, as derived from the Bac*Dive* database. Because of the large number of data points, we chose kernel density estimation (KDE) for visualizing the distribution of observations in the dataset, analogous to a histogram. In contrast to a histogram, a KDE plot smoothes the observations with a Gaussian kernel, producing a continuous density estimate rather than using discrete bins. When applying this to a bivariate dataset, the contours (bands) represent the density. To calculate the labels of the contours, the count of all points within the area is divided by the total area of the contour in the plot.

## Reporting summary
Further information on research design is available in the Nature Portfolio Reporting Summary linked to this article.

## Data availability
The phenotypic training data and associated genome accessions are available under the open CC BY 4.0 license via the BacDive web page[11] and through the BacDive API web service[23]. Predicted phenotypic data can also be accessed via the API using the parameter predictions=1. All models and the corresponding training dataset used in this study are additionally archived and openly available via Zenodo: https://doi.org/10.5281/zenodo.15075932.

## Code availability
The trained machine learning models, as well as a Python application to apply them to other datasets, are available in the following GitHub repository: LeibnizDSMZ/bacdive-AI. The repository includes documentation and usage examples to ensure reproducibility and ease of reuse.

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

## Acknowledgements
We thank our colleague Joaquim Sardá Carbasse for his assistance with integrating the predicted data into the BacDive web page. We also acknowledge funding from the Leibniz Association (SAW project DiASPora: K280/2019).

## Author contributions
Conceptualization: J.K., L.C.R., J.O. Methodology: J.K., L.C.R. Data Curation: L.C.R. Formal Analysis: J.K. Software: J.K. Funding Acquisition: J.O. Investigation: R.P. Visualization: J.K. Supervision: L.C.R., J.O. Writing—original draft: J.K., J.O. Writing—review and editing: J.K., L.C.R., J.O.

## Funding

## Competing interests
The authors declare no competing interests.
