## [Transparent Peer Review file · Communications Biology]

Predicting bacterial phenotypic traits through improved machine learning using high-quality, curated datasets

Corresponding Author: Dr Julia Koblitz

This manuscript has been previously submitted at another journal. This document only contains information relating to versions considered at Communications Biology.

Version 0:

Reviewer comments:

Reviewer #1

(Remarks to the Author)

With the exponentially increasing availability of genomic assemblies from various environments there is also an increasing need in tools that can predict phenotypes from genomic data. The authors present a set of Random Forest classifiers for some major phenotype classes based on gene and domain absence/presence in the BacDive database. The manuscript is generally well-crafted and sufficient care was taken during model construction and testing. The use of some more state-of-the-art models without the need for prior gene prediction and domain annotation might make it a bit more generalizable. Overall, I did not observe any glaring issues and am confident that my comments can be addressed by the authors.

All trained models heavily rely on the prodigal and PFAM domain predictions and I would have liked to see this documented and discussed a bit deeper in the manuscript. Binary PFAM presence/absence patterns were the primary input for the Random Forest models which has some limitations such as lacking PFAM identification in some organisms. Biases when comparing full length reference genomes with metagenomic assemblies and the omission of structural variants such as copy number variations or strong sequence divergence.

It might have been interesting to investigate the relationship between phylogenetic distance and classification performance. For instance, if one would keep an entire phylum as the test set would one still observe good performance? This would help to judge how generalizable the predictions are to novel strains.

I do believe that it might have been interesting to employ some of the current AI architectures such as genomic sequence transformer models and associated foundational models (like described in <https://doi.org/10.1038/s41592-024-02523-z> as an example). This would make the classification independent of de novo gene prediction and PFAM annotation and has the potential to use more of the available sequence information. However, I do understand that this may be out of scope and would leave this up to the authors.

As a minor issue, even though the hyperparameters for the models are provided (#trees, depth, etc.) those lack description of how those were chosen. If model performance was used to choose those this should be mentioned.

Reviewer #2

(Remarks to the Author)

The study aims to enhance the prediction of prokaryotic phenotypic traits from genomic data by leveraging high-quality and standardized datasets from the BacDive database. It evaluates the potential of machine learning (ML) models to infer traits such as oxygen tolerance, motility, and growth temperature, contributing over 50,000 new data points to BacDive.

The authors used curated genomic and phenotypic data from BacDive, sticking to the ensemble learning method Random Forest as the primary method due to its interpretability and robustness

with diverse data, using Protein family annotations (Pfams) as features, and employed several evaluation metrics (F1 score, precision, recall, and normalized Matthews Correlation Coefficient) along with cross-validation, followed by biological interpretation of feature importance. The results are several ML models with high F1 scores for different traits: oxygen requirements, motility, and growth temperature, and provide new biological insights, especially in terms of the link between specific Pfams and growth temperature classifications.

In our opinion, although not a fundamental ML advance, the results represent a significant advance in the application of ML to microbial phenotypes due to the diversity of traits considered and augmentation of existing databases along with biological insights. Our comments follow.

Major Comments

1. The study lacks sufficient context on existing machine learning approaches for phenotype prediction. Also more context about the use of supervised vs unsupervised methods would be useful to better contextualise the specific aims and methods. The reliance on Random Forest, while interpretable, does limit performance compared to ensemble approaches incorporating neural networks or boosting algorithms - more discussion of this issue is warranted.
2. High imbalance in some datasets (e.g., pH optima) reduces model generalizability, as seen in the ACIDO model, which the authors should address more directly. Models like THERMO also showed signs of overfitting at higher temperatures due to limited data, which should be discussed in more detail. Multi-label and continuous data (esp. oxygen requirements) also showed uneven class distribution, possibly undermining generalizability. Needs to be discussed more.
3. Some methodological choices need further clarification / rationale:
 - Line 545-546: Are the two parameters the same for all the models? Need more explanation and justification of the choices of these two hyperparameter settings
 - Line 407-410: The input data is highly imbalanced, so how to reduce the risk of overfitting?
 - Threshold choices (e.g., confidence of 0.5 for positive predictions) may not be optimal for all traits, leading to potential false negatives or positives. Overall, binary classifications (e.g., thermophily) rely heavily on somewhat arbitrarily-chosen thresholds.
4. The final conclusion starting on line 492: "Approach" is a vague term, and this is a complex set of analyses. Can the authors make more specific suggestions for further work / development?
5. Definitions of metrics like normMCC could be expanded or clarified in the methods. On the other hand, most of the others are standard metrics and the authors could present more succinct summaries and refer to appropriate literature.

Minor Comments

1. "Pfams" is not a commonly used machine learning buzzword on its own, but something known mainly to computational biologists in bioinformatics.
2. Abstract (Line 10): "are published openly" → Consider "have been published openly."
3. Introduction (Line 5): "genomic data often fall short of providing functional information since many genes present..." → Consider "as many genes..."
4. Introduction (Line 12): "Genome sequencing has become everyday practice..." → "a routine practice."
5. Results and Discussion (Line 8): "thermophiles" → Define explicitly for broader audience.
6. Abstract, Line 3: Replace "discuss the evaluation metrics" with "discusses evaluation metrics."
7. Introduction, Line 20: Sentence beginning with "Genome sequences provide the basis..." is overly long; consider splitting for clarity.
8. Results, Line 15: "Pfam annotation" should be "Pfam annotations."
9. Introduction, Line 24: "Gram staining" should be "Gram-staining."
10. Table 2 Header: "Description" column aligns poorly in PDF version; check formatting.
11. Conclusion, Line 4: Replace "approach could be advanced" with "approach can be advanced."
12. Line 134: "...as the temperature dependence of growth." - provide citations; e.g., (caveat - a paper from our lab!) (e.g. Smith et al. 2019). And then again in lines 281-283; Smith et al identified the mesophile-thermophile temperature threshold more precisely and objectively.
13. Line 480 Grammar: "Our approach also allows to explicitly..."
14. Figures 2 and 3 are informative but dense. Simplified captions or additional annotations could improve clarity. Label y-axis in panel A of Fig 2.
15. The Introduction and Results sections contain overly long paragraphs; consider breaking them into smaller, focused sections.
16. Supplementary Materials: better citation in main results would help.
17. Paragraph on lines 339-359 is particularly dense and hard to parse
18. Lines 513-514: Clarify what "best annotated" means
19. Lines 518-519: Clarify this - "(except for genomes with a TaxID that matched only on the species level)".

References

1. Smith, Thomas P., Thomas J. H. Thomas, Bernardo Garcia-Carreras, Sofia Sal, Gabriel Yvon-Durocher, Thomas Bell, and Samraat Pawar. 2019. "Community-level respiration of prokaryotic microbes may rise with global warming." *Nature Communications* 10 (1): 5124.

Reviewer #3

(Remarks to the Author)

Koblitz et al. present an interesting machine learning analysis aiming to predict bacterial phenotypes from interpro gene annotation content, based genomes and phenotypic observations present in the BacDive database. In particular, the manuscript outlines prediction results for 8 models including oxygen requirements (aerobe and anaerobe), optimal temperature (thermophile and psychrophile), motility, acidic pH tolerance, gram positive status, and spore formation. Overall, the manuscript is well-written although there are a few small typos and unclear phrasings as pointed out in the minor comments below. This work is original and the results are of interest to the wider community; therefore I would recommend accepting this manuscript after addressing some of the comments below.

Francisco Zorrilla, Institute of Microbiology, ETH Zurich.

Major comments:

- * Considering that most, if not all, models were trained on imbalanced datasets (e.g. 2a, 3b, 4b), then shouldn't the area under the precision recall curve (AUPR) be used for evaluation throughout the text and figures rather than AUC?
- * Why just compare prokka and interpro? Was there any benchmarking with other state of the art gene annotation tools, e.g. eggNOG? I don't expect the entire analysis to be re-run with alternative gene annotation software, but there should be some justification for the tool choice, or perhaps some small benchmarking showing the models are robust to choice of gene annotation tool.
- * Figure 1, 2a, 4a shows that a lot more datapoints could become available for ML with more sequencing of genomes, could this be low hanging fruit to increase training dataset size? Similarly supplementary figure 1 shows that most strains have not been sequenced; in the future, should there be more emphasis placed on sequencing rather than experimental assaying of these strains, especially considering claims made by the authors (lines 52-53) that sequencing is less time consuming/costly than assaying? Although it would increase the quality of the work, I don't expect the authors to carry out further sequencing as part of this work, but perhaps add a sentence or two on this point? e.g. are there plans to sequence the remaining strains?
- * Are the training data files available online? It looks like the data is available through the BacDive API, but there is no way to easily download all the data in a single file from what I could see. Could these data be uploaded to the GitHub repo as tsv/csv files? This would make the analysis more easily reproduced
- * It would be interesting to the wider community to discuss briefly how this work relates and compares to genomeSPOT, which also uses BacDive data but takes a different approach in the features used for model trainings (Predicting microbial growth conditions from amino acid composition, Barnum et al. 2024, <https://www.biorxiv.org/content/10.1101/2024.03.22.586313v1>).
- * It is not strictly necessary, but I think that further experimental validation would really strengthen the manuscript. For example, take 1-10 microbes with genome sequences but no recorded phenotypic traits, then carry out the experimental assays to evaluate if the model predictions are accurate. Alternatively, or in addition, it would be interesting to take species that have not been sequenced but have been experimentally assayed, sequence them, annotate genes, and then check if the model-based predictions agree with the experimental observations
- * Similar to the above comment, this is not strictly required, but it would further strengthen the manuscript if the machine learning models were applied to uncultured microbes that have been sequenced (i.e. MAGs) to predict growth conditions, then check if it can be cultivated under the predicted conditions. This would really showcase the power of the models.

Minor comments:

- * Line 299 "little satisfactory" unclear idiosyncratic language -> perhaps rephrase as "unsatisfactory"?
- * Figure 4a, somewhat unclear what hatched bars represent in panel A. Does it mean that for model motile_0 gliding was considered motile but for model motile_2 it was considered non-motile?
- * Line 340 "except of one that is part of" -> "except for one that is part of"
- * Figure 5b, what does the arrow represent in each plate? Clarify that this represent motile bacteria?
- * Line 421 "metrices" -> "metrics"
- * Line 422 explain what NVP stands for
- * Line 427 "as the normMCC" -> "as the normMCC metric"
- * Line 475 "completely new datasets for" -> "completely new datapoints for ". Same comment for caption of supplementary figure 3
- * Line 477 "biologically deep machine learning models", what does biologically deep mean? Did you mean "biologically related" or "microbiological" deep learning models?

Version 1:

Reviewer comments:

Reviewer #1

(Remarks to the Author)

The authors have addressed most of my comments and I believe the manuscript has improved significantly.

As a minor comment, I would disagree with the new statement added in lines 633–640. Random Forest models encode no underlying biology, and feature importances calculated for Random Forest models are not that different from the ones that can be calculated for XGBOOST (or SHAP values for other models). They are all unrelated to any biological mechanism or causality. So stating that Random Forest models are in any view more biologically relevant than boosted trees is not correct in my view.

Apart from this, I have no further concerns at this point.

Reviewer #3

(Remarks to the Author)

Thank you for your thorough revisions. I appreciate the effort you put into addressing reviewer comments, and I'm pleased with the improvements made. The paper is much clearer now, and I have no further concerns. Well done!

Reviewer #1

With the exponentially increasing availability of genomic assemblies from various environments there is also an increasing need in tools that can predict phenotypes from genomic data. The authors present a set of Random Forest classifiers for some major phenotype classes based on gene and domain absence/presence in the BacDive database. The manuscript is generally well-crafted and sufficient care was taken during model construction and testing. The use of some more state-of-the-art models without the need for prior gene prediction and domain annotation might make it a bit more generalizable. Overall, I did not observe any glaring issues and am confident that my comments can be addressed by the authors.

We are grateful for the reviewer's thoughtful and constructive feedback. Please find our detailed responses to each of the comments below.

- All trained models heavily rely on the prodigal and PFAM domain predictions and I would have liked to see this documented and discussed a bit deeper in the manuscript. Binary PFAM presence/absence patterns were the primary input for the Random Forest models which has some limitations such as lacking PFAM identification in some organisms. Biases when comparing full length reference genomes with metagenomic assemblies and the omission of structural variants such as copy number variations or strong sequence divergence.
 - We compared a number of different annotation methods and found Pfam domains to be most reliable, meaningful to analyse and versatile for this approach. We added a new section to the results and discussion to highlight this a bit further (please see **Lines 141ff**).
- It might have been interesting to investigate the relationship between phylogenetic distance and classification performance. For instance, if one would keep an entire phylum as the test set would one still observe good performance? This would help to judge how generalizable the predictions are to novel strains.
 - We thank the reviewer for this thoughtful suggestion. We agree that assessing model generalizability across phylogenetic distances is an important aspect of evaluating predictive robustness. To address this point, we conducted an additional analysis comparing classification performance and prediction confidence across all bacterial and archaeal phyla (see new Figure 6). Our results show that while classification accuracy remains generally high across most phyla (median: 94.9%), prediction confidence varies more strongly, particularly in phyla that are metabolically or ecologically distinct, such as Chloroflexota and Deinococcota. Notably, underrepresented phyla with small sample sizes also exhibit more variable performance, though some smaller phyla still achieve robust predictions. We have added an additional section to the manuscript to discuss these observations (**Line 421ff**). We hope this analysis addresses the reviewer's concern and strengthens the manuscript.
- I do believe that it might have been interesting to employ some of the current AI architectures such as genomic sequence transformer models and associated foundational

models (like described in <https://doi.org/10.1038/s41592-024-02523-z> as an example). This would make the classification independent of de novo gene prediction and PFAM annotation and has the potential to use more of the available sequence information. However, I do understand that this may be out of scope and would leave this up to the authors.

- We looked into this topic as well and consulted other people working on this. The consensus was that in most of the cases genomic sequence transformer models are, because of their nature, very sensitive to irrelevant patterns in the genome. They easily learn on SNPs or non-coding parts of the DNA, making them even more prone to taxonomic bias. It was shown by our colleagues that traits, which are very dependent on the taxonomic classification of a strain (e.g. Gram staining behavior), can be easily predicted by such models with high confidence. However, the models often fail to predict traits that are spread among taxonomic groups even though they might just need a couple of genes to be present, such as oxygen requirement. Because of this we decided to rely on a classical feature-based machine learning method even though they might be less reliable for strains from less-studies taxonomic groups. We added a paragraph to the Results and Discussion (**Lines 160ff**) to emphasize this in the manuscript.
- As a minor issue, even though the hyperparameters for the models are provided (#trees, depth, etc.) those lack description of how those were chosen. If model performance was used to chose those this should be mentioned.
 - Thank you for pointing out the need for clarification regarding our hyperparameter and threshold choices. We have clarified these points in the revised Materials & Methods section of our manuscript (**Lines 641ff**).

Reviewer #2

The study aims to enhance the prediction of prokaryotic phenotypic traits from genomic data by leveraging high-quality and standardized datasets from the BacDive database. It evaluates the potential of machine learning (ML) models to infer traits such as oxygen tolerance, motility, and growth temperature, contributing over 50,000 new data points to BacDive.

The authors used curated genomic and phenotypic data from BacDive, sticking to the ensemble learning method Random Forest as the primary method due to its interpretability and robustness with diverse data, using Protein family annotations (Pfams) as features, and employed several evaluation metrics (F1 score, precision, recall, and normalized Matthews Correlation Coefficient) along with cross-validation, followed by biological interpretation of feature importance. The results are several ML models with high F1 scores for different traits: oxygen requirements, motility, and growth temperature, and provide new biological insights, especially in terms of the link between specific Pfams and growth temperature classifications.

In our opinion, although not a fundamental ML advance, the results represent a significant advance in the application of ML to microbial phenotypes due to the diversity of traits considered and augmentation of existing databases along with biological insights. Our comments follow.

We appreciate the reviewer's careful reading of our manuscript and the valuable suggestions provided. Below, we address each comment individually.

Major Comments

- The study lacks sufficient context on existing machine learning approaches for phenotype prediction. Also more context about the use of supervised vs unsupervised methods would be useful to better contextualise the specific aims and methods. The reliance on Random Forest, while interpretable, does limit performance compared to ensemble approaches incorporating neural networks or boosting algorithms - more discussion of this issue is warranted.
 - Thank you for your comment. We added a more detailed review to the introduction (**Lines 99ff**) and to the materials and methods (**Lines 633ff**) to justify our choice better.
- High imbalance in some datasets (e.g., pH optima) reduces model generalizability, as seen in the ACIDO model, which the authors should address more directly. Models like THERMO also showed signs of overfitting at higher temperatures due to limited data, which should be discussed in more detail. Multi-label and continuous data (esp. oxygen requirements) also showed uneven class distribution, possibly undermining generalizability. Needs to be discussed more.
 - We understand the confusion, but the imbalanced datasets have been specifically included into this part of the discussion to demonstrate the problems if accuracy is used as a single metric. Even though the data is highly imbalanced, the accuracy determined is still outstanding. Since the latter metric thus is not sufficient which is why we recommend to include several other metrics into published models. We rephrased this section a bit to make this point more clear to the audience (**see line 483f**).
- Some methodological choices need further clarification / rationale:
 - Line 545-546: Are the two parameters the same for all the models? Need more explanation and justification of the choices of these two hyperparameter settings
 - Thank you for pointing out the need for clarification regarding our hyperparameter and threshold choices. We have clarified these points in the revised Materials & Methods section of our manuscript (**Lines 641ff**).
 - Line 407-410: The input data is highly imbalanced, so how to reduce the risk of overfitting?
 - Please see our answer to the previous major point above regarding the imbalance of the dataset.
 - Threshold choices (e.g., confidence of 0.5 for positive predictions) may not be optimal for all traits, leading to potential false negatives or positives. Overall, binary classifications (e.g., thermophily) rely heavily on somewhat arbitrarily-chosen thresholds.
 - Thank you for highlighting this point. Regarding your first comment on threshold choices, we have expanded the description in the hyperparameter section to provide additional context and clarify our approach (see **Lines 747f**). Concerning your second point, we agree that thresholds significantly impact binary classifications. Specifically for the thermophily dataset, we have extensively discussed the selection

and implications of classification thresholds in the manuscript, dedicating a substantial part of the relevant section to addressing precisely this issue. We have added more citations to make this point even more clear now in the manuscript **(Line 283)**

- The final conclusion starting on line 492: "Approach" is a vague term, and this is a complex set of analyses. Can the authors make more specific suggestions for further work / development?
 - Thank you for highlighting this. We have revised the conclusion to provide specific directions for future development, including systematic expansion of genomic datasets through targeted sequencing, performing laboratory validation experiments to verify predictions, and applying the developed models to predict cultivation conditions for uncultured organisms **(Lines 571ff)**. We believe these clear suggestions substantially improve the outlook provided by the manuscript.
- Definitions of metrics like normMCC could be expanded or clarified in the methods. On the other hand, most of the others are standard metrics and the authors could present more succinct summaries and refer to appropriate literature.
 - We included this thorough explanation of standard metrics because we found this article to be also relevant to microbiologists who might not be that knowledgeable about the metrics. However, we shortened the explanation on standard metrics and added more information on the transformation of the MCC to normMCC as suggested **(Line 702f)**: "To facilitate comparability with other metrics reported on a [0, 1] scale, we linearly rescaled the MCC using min-max normalization and refer to this transformed score as *normMCC*."

Minor Comments

- We thank the reviewer for language editing, which we have incorporated into the manuscript if not stated otherwise.
- "Pfams" is not a commonly used machine learning buzzword on its own, but something known mainly to computational biologists in bioinformatics.
 - We replaced Pfams with protein families where possible.
- Abstract (Line 10): "are published openly" → Consider "have been published openly."
- Introduction (Line 5): "genomic data often fall short of providing functional information since many genes present..." → Consider "as many genes..."
- Introduction (Line 12): "Genome sequencing has become everyday practice..." → "a routine practice."
- Results and Discussion (Line 8): "thermophiles" → Define explicitly for broader audience.
 - We have defined this term and provided references.
- Abstract, Line 3: Replace "discuss the evaluation metrics" with "discusses evaluation metrics."
- Introduction, Line 20: Sentence beginning with "Genome sequences provide the basis..." is overly long; consider splitting for clarity.

- Results, Line 15: "Pfam annotation" should be "Pfam annotations."
- Introduction, Line 24: "Gram staining" should be "Gram-staining."
- Table 2 Header: "Description" column aligns poorly in PDF version; check formatting.
- Conclusion, Line 4: Replace "approach could be advanced" with "approach can be advanced."
- Line 134: "...as the temperature dependence of growth. " - provide citations; e.g., (caveat - a paper from our lab!) (e.g, Smith et al. 2019). And then again in lines 281-283; Smith et al identified the mesophile-thermophile temperature threshold more precisely and objectively.
 - We incorporated the suggested reference in another chapter where we found it more appropriate. It can be found as reference 35 in line 312.
- Line 480 Grammar: "Our approach also allows to explicitly..."
- Figures 2 and 3 are informative but dense. Simplified captions or additional annotations could improve clarity. Label y-axis in panel A of Fig 2.
 - It is in our opinion unusual to add figure titles to scientific articles. We have added a label to the x-axis.
- The Introduction and Results sections contain overly long paragraphs; consider breaking them into smaller, focused sections.
 - We have carefully reviewed the structure of the Introduction and Results sections and revised several paragraphs to improve readability and focus.
- Supplementary Materials: better citation in main results would help.
 - We ensured to improve the citation of supplementary data in the main manuscript.
- Paragraph on lines 339-359 is particularly dense and hard to parse
- Lines 513-514: Clarify what "best annotated" means
- Lines 518-519: Clarify this - "(except for genomes with a TaxID that matched only on the species level)".

Reviewer #3

Koblitz et al. present an interesting machine learning analysis aiming to predict bacterial phenotypes from interpro gene annotation content, based genomes and phenotypic observations present in the BacDive database. In particular, the manuscript outlines prediction results for 8 models including oxygen requirements (aerobe and anaerobe), optimal temperature (thermophile and psychrophile), motility, acidic pH tolerance, gram positive status, and spore formation. Overall, the manuscript is well-written although there are a few small typos and unclear phrasings as pointed out in the minor comments below. This work is original and the results are of interest to the wider community; therefore I would recommend accepting this manuscript after addressing some of the comments below.

We thank the reviewer for taking the time to revise our manuscript and for the constructive feedback. We reply one comment at a time below.

Major comments:

- Considering that most, if not all, models were trained on imbalanced datasets (e.g. 2a, 3b, 4b), then shouldn't the area under the precision recall curve (AUPR) be used for evaluation throughout the text and figures rather than AUC?
 - Thank you for this helpful suggestion. In our initial analysis, we primarily reported AUC (area under the receiver operating characteristic curve) because it is widely used in classification tasks and provides an intuitive way to compare model performance across different datasets. However, we agree that AUPR could provide additional insights into the predictive performance, especially for datasets with class imbalances.
 - To address this, we have now computed AUPR for all models and included the values in **figure 4 and supplementary figure S2**. Additionally, we have updated the text in the results and discussion to clarify how AUPR better reflects the model's ability to distinguish the positive class in imbalanced datasets.
- Why just compare prokka and interpro? Was there any benchmarking with other state of the art gene annotation tools, e.g. eggNOG? I don't expect the entire analysis to be re-run with alternative gene annotation software, but there should be some justification for the tool choice, or perhaps some small benchmarking showing the models are robust to choice of gene annotation tool.
 - We indeed did a larger benchmarking comparing a number of different annotation methods and found Pfam domains to be most reliable, meaningful to analyse and versatile for this approach. We added a new section named „Evaluation and justification of protein annotation methods and modeling approaches“ to the discussion in the manuscript to explain this a bit further. Please see **Lines 141ff**.
- Figure 1,2a,4a shows that a lot more datapoints could become available for ML with more sequencing of genomes, could this be low hanging fruit to increase training dataset size? Similarly supplementary figure 1 shows that most strains have not been sequenced; in the future, should there be more emphasis placed on sequencing rather than experimental assaying of these strains, especially considering claims made by the authors (lines 52-53) that sequencing is less time consuming/costly than assaying? Although it would increase the quality of the work, I don't expect the authors to carry out further sequencing as part of this work, but perhaps add a sentence or two on this point? e.g. are there plans to sequence the remaining strains?
 - We fully agree with the reviewer that expanding genomic datasets represents a valuable and practical step toward increasing training dataset size and enhancing prediction quality. To address this suggestion, we have explicitly emphasized the importance of targeted sequencing initiatives within culture collections in the revised perspectives section (**Line 571ff**). However, while specific sequencing plans exist within our institute, these efforts lie beyond the scope of this manuscript.
- Are the training data files available online? It looks like the data is available through the BacDive API, but there is no way to easily download all the data in a single file from what I could see. Could these data be uploaded to the GitHub repo as tsv/csv files? This would make the analysis more easily reproduced.

- Thank you for this comment. We uploaded the training data to Zenodo as requested: <https://doi.org/10.5281/zenodo.15075932> (files were too large for GitHub) and added this information to the section `Data and code availability` (**Line 755**).
- It would be interesting to the wider community to discuss briefly how this work relates and compares to genomeSPOT, which also uses BacDive data but takes a different approach in the features used for model trainings (Predicting microbial growth conditions from amino acid composition, Barnum et al. 2024, [biorXiv https://www.biorxiv.org/content/10.1101/2024.03.22.586313v1](https://www.biorxiv.org/content/10.1101/2024.03.22.586313v1)).
 - In this study, we limit the predictions to phenotypic traits. Although the aforementioned study is interesting, we are afraid that the scope is totally different.
- It is not strictly necessary, but I think that further experimental validation would really strengthen the manuscript. For example, take 1-10 microbes with genome sequences but no recorded phenotypic traits, then carry out the experimental assays to evaluate if the model predictions are accurate. Alternatively, or in addition, it would be interesting to take species that have not been sequenced but have been experimentally assayed, sequence them, annotate genes, and then check if the model-based predictions agree with the experimental observations
 - We appreciate this valuable suggestion for additional experimental validation. While targeted laboratory validation is beyond the scope of our current study due to practical constraints, we recognize its critical role in confirming the accuracy and reliability of our predictions. To reflect this, we have expanded the outlook section of the manuscript, explicitly outlining possible validation strategies and emphasizing their importance for future research. Please see **lines 578ff** for this.
- Similar to the above comment, this is not strictly required, but it would further strengthen the manuscript if the machine learning models were applied to uncultured microbes that have been sequenced (i.e. MAGs) to predict growth conditions, then check if it can be cultivated under the predicted conditions. This would really showcase the power of the models.
 - We agree that demonstrating the applicability of our models to uncultivated microbes and MAGs would significantly showcase their practical power. However, predicting growth conditions solely from physicochemical traits, as currently performed by our models, addresses only part of the complexity needed for successful cultivation. Thus, we explicitly discussed this limitation in the revised manuscript and mentioned ongoing complementary work aimed at predicting cultivation media components, based on additional databases such as *MediaDive* (**Lines 582f**).

Minor comments:

- Line 299 “little satisfactory” unclear idiosyncratic language -> perhaps rephrase as “unsatisfactory”?
 - **Rephrased.**
- Figure 4a, somewhat unclear what hatched bars represent in panel A. Does it mean that for model motile_0 gliding was considered motile but for model motile_2 it was considered non-motile?

- The schema is the same as in panel B, it refers to which datasets were used as negative (hatched) and positive (filled) labels. We added a statement to the figure description to clarify that.
- Line 340 “except of one that is part of” -> “except for one that is part of”
 - Rephrased.
- Figure 5b, what does the arrow represent in each plate? Clarify that this represent motile bacteria?
 - We have added the following paragraph to the caption of figure 5b: „The arrows point to the edge of the opaque area to which the cells have moved.“
- Line 421 “metrices” -> “metrics”
 - Changed.
- Line 422 explain what NVP stands for
 - We apologize for the typo. We added the meaning of NPV to the legend of table 4 and to the text: NPV = negative predictive value.
- Line 427 “as the normMCC” -> “as the normMCC metric”
 - Rephrased.
- Line 475 “completely new datasets for” -> “completely new datapoints for “. Same comment for caption of supplementary figure 3
 - Rephrased.
- Line 477 “biologically deep machine learning models”, what does biologically deep mean? Did you mean “biologically related” or “microbiological” deep learning models?
 - We changed that into „machine learning models that take biological insights into account“.

Changes to Figures:

Figure 2: Panel A: y-axis has been labeled.

Figure 4: Panel C has been changed from a ROC to a Precision/Recall Curve.

Figure 6: completely new figure

Reviewer #1

The authors have addressed most of my comments and I believe the manuscript has improved significantly.

We like to thank the reviewer for providing their expertise and taking the time to improve our manuscript. It was highly appreciated.

As a minor comment, I would disagree with the new statement added in lines 633–640. Random Forest models encode no underlying biology, and feature importances calculated for Random Forest models are not that different from the ones that can be calculated for XGBOOST (or SHAP values for other models). They are all unrelated to any biological mechanism or causality. So stating that Random Forest models are in any view more biologically relevant than boosted trees is not correct in my view.

We apologize for this; it was not our intention to make such false statements. We got this mixed up when answering comments about deep learning algorithms and gradient boosting. We rephrased the paragraph to transport what we intended to say: that deep learning algorithms are not as easy to interpret as supervised machine learning approaches. We further added a statement implying that other algorithms would have been suitable for this as well, including support vector machines and gradient boosting algorithms. We sincerely hope that the statement now reflects the position of the reviewer.

Apart from this, I have no further concerns at this point.

Thanks again for taking the time.

Reviewer #3

Thank you for your thorough revisions. I appreciate the effort you put into addressing reviewer comments, and I'm pleased with the improvements made. The paper is much clearer now, and I have no further concerns. Well done!

We would like to thank the reviewer for taking their time to review our manuscript. We valued your input and agree that it improved the quality of our paper. Thank you!

The study aims to enhance the prediction of prokaryotic phenotypic traits from genomic data by leveraging high-quality and standardized datasets from the BacDive database. It evaluates the potential of machine learning (ML) models to infer traits such as oxygen tolerance, motility, and growth temperature, contributing over 50,000 new data points to BacDive.

The authors used curated genomic and phenotypic data from BacDive, sticking to the ensemble learning method Random Forest as the primary method due to its interpretability and robustness with diverse data, using Protein family annotations (Pfam) as features, and employed several evaluation metrics (F1 score, precision, recall, and normalized Matthews Correlation Coefficient) along with cross-validation, followed by biological interpretation of feature importance. The results are several ML models with high F1 scores for different traits: oxygen requirements, motility, and growth temperature, and provide new biological insights, especially in terms of the link between specific Pfams and growth temperature classifications.

In our opinion, although not a fundamental ML advance, the results represent a significant advance in the application of ML to microbial phenotypes due to the diversity of traits considered and augmentation of existing databases along with biological insights. Our comments follow.

Major Comments

1. The study lacks sufficient context on existing machine learning approaches for phenotype prediction. Also more context about the use of supervised vs unsupervised methods would be useful to better contextualise the specific aims and methods. The reliance on Random Forest, while interpretable, does limit performance compared to ensemble approaches incorporating neural networks or boosting algorithms - more discussion of this issue is warranted.
2. High imbalance in some datasets (e.g., pH optima) reduces model generalizability, as seen in the ACIDO model, which the authors should address more directly. Models like THERMO also showed signs of overfitting at higher temperatures due to limited data, which should be discussed in more detail. Multi-label and continuous data (esp. oxygen requirements) also showed uneven class distribution, possibly undermining generalizability. Needs to be discussed more.
3. Some methodological choices need further clarification / rationale:
 - Line 545-546: Are the two parameters the same for all the models? Need more explanation and justification of the choices of these two hyperparameter settings
 - Line 407-410: The input data is highly imbalanced, so how to reduce the risk of overfitting?
 - Threshold choices (e.g., confidence of 0.5 for positive predictions) may not be optimal for all traits, leading to potential false negatives or positives. Overall, binary classifications (e.g., thermophily) rely heavily on somewhat arbitrarily-chosen thresholds.
4. The final conclusion starting on line 492: Approach" is a vague term, and this is a complex set of analyses. Can the authors make more specific suggestions for further work / development?
5. Definitions of metrics like normMCC could be expanded or clarified in the methods. On the other hand, most of the others are standard metrics and the authors could present more succinct summaries and refer to appropriate literature.

Minor Comments

1. "Pfam" is not a commonly used machine learning buzzword on its own, but something known mainly to computational biologists in bioinformatics.
2. Abstract (Line 10): "are published openly" → Consider "have been published openly."
3. Introduction (Line 5): "genomic data often fall short of providing functional information since many genes present..." → Consider "as many genes..."
4. Introduction (Line 12): "Genome sequencing has become everyday practice..." → "a routine practice."
5. Results and Discussion (Line 8): "thermophiles" → Define explicitly for broader audience.
6. Abstract, Line 3: Replace "discuss the evaluation metrics" with "discusses evaluation metrics."
7. Introduction, Line 20: Sentence beginning with "Genome sequences provide the basis..." is overly long; consider splitting for clarity.
8. Results, Line 15: "Pfam annotation" should be "Pfam annotations."
9. Introduction, Line 24: "Gram staining" should be "Gram-staining."
10. Table 2 Header: "Description" column aligns poorly in PDF version; check formatting.
11. Conclusion, Line 4: Replace "approach could be advanced" with "approach can be advanced."
12. Line 134: "...as the temperature dependence of growth. " - provide citations; e.g., (caveat - a paper from our lab!) (e.g, Smith et al. 2019). And then again in lines 281-283; Smith et al identified the mesophile-thermophile temperature threshold more precisely and objectively.
13. Line 480 Grammar: "Our approach also allows to explicitly..."
14. Figures 2 and 3 are informative but dense. Simplified captions or additional annotations could improve clarity. Label y-axis in panel A of Fig 2.
15. The Introduction and Results sections contain overly long paragraphs; consider breaking them into smaller, focused sections.
16. Supplementary Materials: better citation in main results would help.
17. Paragraph on lines 339-359 is particularly dense and hard to parse
18. Lines 513-514: Clarify what "best annotated" means
19. Lines 518-519: Clarify this - "(except for genomes with a TaxID that matched only on the species level)".

References

1. Smith, Thomas P., Thomas J. H. Thomas, Bernardo Garcia-Carreras, Sofia Sal, Gabriel Yvon-Durocher, Thomas Bell, and Samraat Pawar. 2019. "Community-level respiration of prokaryotic microbes may rise with global warming." *Nature Communications* 10 (1): 5124.

Samraat Pawar, Zhongbin Hu and Yan Zu
Silwood Park, Imperial College London

21 Dec 2024